# The Weddell Gyre heat budget associated with the Warm Deep Water circulation derived from Argo Floats

Krissy Anne Reeve[1], Torsten Kanzow[1,2], Olaf Boebel[1], Myriel Vredenborg[1], Volker Strass[1], Rüdiger Gerdes[1,3]

[1]Alfred Wegener Institute, Bremerhaven, Germany

[2]Bremen University, Department of Physics and Electrical Engineering, Bremen, Germany

[3]Jacobs University, Bremen, Germany

**Correspondence:** Krissy Anne Reeve (krissy.reeve@awi.de)

**Abstract:** The Weddell Gyre plays an important role in the global climate system by supplying heat to underneath the ice shelves, and to the formation of deep and bottom water masses, which have been subject to widespread warming over past decades. In this study, we investigate the redistribution of heat throughout the Weddell Gyre by diagnosing the terms of the heat conservation equation for a 1000 m thick layer of water encompassing the core of Warm Deep Water. The spatial distribution of the different advective and diffusive terms in terms of heat tendencies are estimated using gridded climatologies of temperature and velocity, obtained from Argo floats in the Weddell Gyre from 2002 to 2016. While the results are somewhat noisy on the grid scale, and the representation of the effects of eddy mixing is highly uncertain due to need to parameterise them by means of turbulent diffusion, the heat budget (i.e., the sum of all terms) closes (within the uncertainty range) when integrated over the open inflow region in the southern limb, whereas the interior circulation cell remains unbalanced. There is an overall balance in the southern limb between the mean horizontal advection and horizontal turbulent diffusion of heat, whereas the vertical terms contribute comparatively little to the heat budget throughout the Weddell Gyre. Heat convergence due to mean horizontal advection balances with divergence due to horizontal turbulent diffusion in the open southern limb of the Weddell Gyre. In contrast, heat divergence due to mean horizontal advection is much weaker than convergence due to horizontal turbulent diffusion in the interior circulation cell of the Weddell Gyre, due to large values in the latter along the northern boundary, due to large meridional temperature gradients. Heat is advected into the Weddell Gyre along the southern limb, some of which is turbulently diffused northwards into the interior circulation cell, while some is likely turbulently diffused southwards towards the shelf seas. This suggests that horizontal turbulent diffusion plays a role in transporting heat both towards the gyre interior where upwelling occurs, as well as towards the ice shelves. Horizontal turbulent diffusion is also a mechanism by which heat can be transported into the Weddell Gyre across the open northern boundary. Temporal deviations from the mean terms are not included due to study limitations. In order to appreciate the role of transient eddying processes, a continued effort to increase the spatial and temporal coverage of observations in the eastern Weddell Sea is required.

**Plain Language Summary:** Ocean currents in the Weddell Sea are governed by a wind-driven clockwise circulating gyre, which is connected to the Antarctic Circumpolar Current to its north. Warm and salty deep water enters the Weddell Sea in its east, and is transported by the gyre circulation first southward, then westward, and back northwards again following the continental boundaries. During this circulation, some of this water becomes lighter by mixing with the overlying surface waters, thus shoaling as it circulates. It also loses heat to the atmosphere and by contact with the ice shelves. When this occurs, the water becomes heavier, as also by salt released during the freezing of sea ice. The heaviest waters sink downwards along the Antarctic continental slope, eventually filling the deep abyssal ocean basins. The main source water mass for these processes and also the main source of heat to the Weddell Sea is called Warm Deep Water. Previous studies have shown the whole water column, especially in the deeper layers, is warming over recent decades in the Weddell Sea. The temperature of Warm Deep Water, however, fluctuates too strongly to tease out long-term trends from the "snapshot" data that is available to

us. To better understand how heat is distributed in the Weddell Gyre within the Warm Deep Water, we combine temperature and velocity observations from a fleet of Argo floats freely drifting throughout the Weddell Gyre between 2002 and 2016. Using these observations, we estimate a heat budget in the layer that extends 1000 m deep from below the surface layer. This layer always includes the core of Warm Deep Water, regardless of its vertical position in the water column. Overall, large uncertainty prevents us from interpreting the results on a local scale, but interpretable features of heat flux divergence and convergence emerge when integrating the heat budget over large areas. The large-scale currents carry heat into the westward-flowing southern limb from the east, and upwelling brings heat upwards from below the layer throughout the whole gyre. Turbulent mixing, representing small scale processes, removes heat from the Warm Deep Water core through the top of the layer upwards into the ocean surface throughout. It also removes heat from the southern limb, northwards into the central gyre where Warm Deep Water recirculates and moves closer to the surface, as well as southwards towards the Antarctic coastline. Lastly, turbulent mixing also brings heat into the gyre across the northern boundary.

# 1    Introduction

Understanding the drivers and pathways of large-scale ocean circulation is a fundamental component of climate science (Rhein et al., 2013). To comprehend the regulatory role of the oceans in the climate system, one can determine the ocean heat budget, which describes the redistribution of heat throughout the ocean by means of horizontal and vertical advection, turbulent diffusion, and surface heat fluxes to the atmosphere (e.g., Tamsitt et al., 2016; an adapted form of the heat budget equation is given in Eq. 1.1).

The Weddell Gyre is located south of 50° S in the Atlantic sector of the Southern Ocean, where Circumpolar Deep Water (CDW) predominantly enters the gyre's southern limb in the east at about 30° E. The CDW that enters the Weddell Gyre becomes modified, and is commonly referred to as Warm Deep Water (WDW). WDW circulates the cyclonic gyre, undergoing cooling and freshening en-route, through interaction with the underlying and overlying water masses (e.g., Fahrbach et al., 2004; Klatt et al., 2005; Fahrbach et al., 2011; Leach et al., 2011). The core of WDW is identified as the sub-surface temperature maximum (hereon referred to as $\Theta_{max}$), which feeds heat into the Weddell Gyre (Fahrbach et al., 2004, 2011; Cisewski et al., 2011; Ryan et al., 2016). The distribution of temperature at the depth of the $\Theta_{max}$ is shown in Fig. 1, which is derived from in situ observations from Argo floats (from Reeve et al., 2016, 2019). The Weddell Gyre has been ascribed the role of a heat buffer (Fahrbach et al., 2011), in that it acts to store and redistribute heat and salt in the water column, effectively transferring heat to the deeper layers where it is ultimately exported northwards, spreading throughout the abyssal global ocean (e.g., Foster et al., 1987; Naveira Garabato et al., 2002, 2016; Fahrbach et al., 2011).

Warming trends over recent decades have been observed in the WSDW and WSBW (Weddell Sea –Deep and -Bottom Water respectively) (Fahrbach et al., 2011; Meredith et al., 2011; Strass et al., 2020). However, their primary source of heat, WDW, exhibits pronounced decadal variations and shows no significant long-term warming trend (Fahrbach et al., 2011; Kerr et al., 2017). While Strass et al. (2020) show that the whole water column below 700 m exhibits a significant long-term warming trend, which would incorporate the lower part of the WDW layer, they also observed significant variability and even areas of cooling in the upper 700 m, which incorporates a significant chunk of the WDW layer.

In this study, we combine observations of the velocity field (Reeve et al., 2019) with the temperature field (Reeve et al., 2016), both derived primarily from Argo floats, to diagnose components of the heat budget of a fixed volume of water fully encompassing the core of WDW within the Weddell Gyre. Given Argo float data lacks the spatial-temporal coverage to resolve the seasonal cycle while also objectively mapping the entire region, a full basin analysis of the upper 50 m is unfeasible. By analysing the heat budget for a fixed volume encompassing the core of WDW, we can, however, determine the ways in which heat from WDW is redistributed throughout the Weddell Gyre. The rest of this paper is structured as follows. Section 2

describes the data sources used in this study, while Section 3 details the method of applying the heat budget, and its associated estimation of uncertainty. Section 4 presents the individual heat budget terms for the Weddell Gyre for the whole region as well as integrated over specific areas, while Section 5 interprets these results in context of the study limitations and the literature. Lastly, Section 6 provides our final summary.

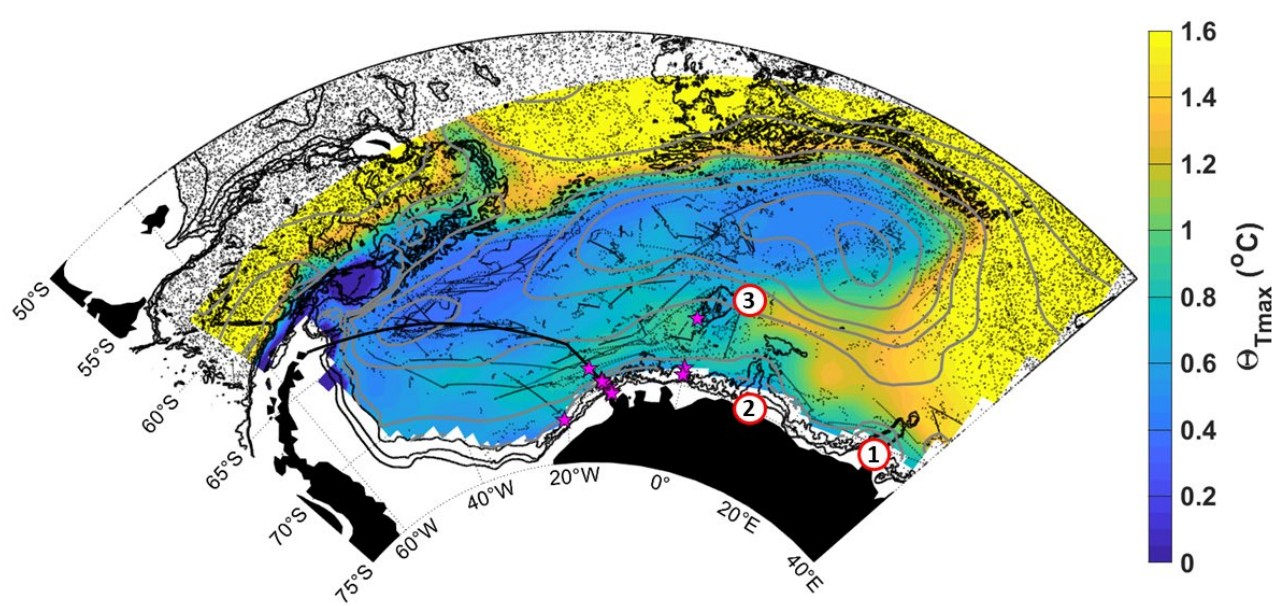

**Figure 1: Sub-surface conservative temperature maximum ($\Theta_{max}$) with streamlines (grey contours) of the vertically integrated stream function for 50-2000 dbar with a spacing of 5 Sv, derived from in-situ observations from Argo floats (Reeve et al., 2019, 2016). Black dots show Argo float profile positions and magenta stars show mooring positions used in velocity field estimates. The thick black line shows the repeat ship-based transect from Kapp Norvegia to Joineville Island. The red circles labelled 1-3 show (1) Gunnerus Ridge, (2) Astrid Ridge and (3) Maud Rise. The black contours show the 1000, 2000 and 3000 m isobaths, from the general bathymetric chart of the oceans (GEBCO, IOC et al., 2003).**

85

## 2    Data sources

For this study we computed the heat budget of the entire Weddell Gyre using different sets of data from various sources, which will be described in more detail in the following subsections.

### 2.1    Data sources

Table 1 describes the data sources used for determining the heat budget presented in this paper. Each source is then described in further detail in Sections 2 and 3, as well as in the Supplement (S1-4).

Table 1. A summary of the associated data source, method and citation for each variable required in the heat budget (Eq. 1.2 in Section 3, and column 2 below).

| Section for details | Variable | Data source | Methodology | Citation |
|---|---|---|---|---|
| 2.2 | Gridded Temperature ($\theta$) | Argo floats (2002-2016) | Objective mapping | Reeve et al., 2016 |
| 2.2 | Gridded horizontal velocity ($U$) | Argo floats (2002-2016) | Objective mapping, cost function | Reeve et al., 2019 |

| 2.2 | Long-term mean velocity at the shelf edge (U) | Moorings (1989-2016) | | Le Piah et al., 2020 |
|---|---|---|---|---|
| 2.2 | Bathymetry | GEBCO | | IOC et al., 2003 |
| S3 | Vertical (Ekman) velocity ($w_E$) | ERA5-interim reanalysis and Polar Pathfinder (listed below) | Regridded using a distance-weighted mean | Hersbach et al., (2020) & Tschudi et al., (2019) |
| S3 | Wind stress ($\tau_{air-ocean}$) derived from wind field at 10 m above sea surface ($U_{10}$) | ERA5-interim reanalysis | Used to compute $w_E$ | Hersbach et al., (2020) |
| S3 | Sea-ice concentration ($\alpha$) | ERA5-interim reanalysis | Used to compute $w_E$ | Hersbach et al., (2020) |
| S3 | Sea-ice velocity ($U_{ice}$) | Polar Pathfinder Daily 25 km EASE-Grid Sea Ice Motion Vectors | Used to compute $w_E$ | Tschudi et al., (2019) |
| 3 & S4 | Horizontal diffusivity ($\kappa_H$) | Sevellec et al., (2022), and estimates in the literature (see citation) | Where available (i.e., outskirts of gyre) the Sevellec dataset. Infilled within gyre with a constant value based on Donnelly et al. 2017 | Sevellec et al., 2022; Donnelly et al. (2017) |
| 3 | Vertical diffusivity ($\kappa_V$) | Based on estimates in the literature (see citation) | | Donnelly et al. (2017) |
| S2 | Air-sea heat fluxes | ERA-interim reanalysis | | Jones et al., 2016 |

## 2.2     Gridded velocity and temperature fields derived from Argo floats:

All Argo float data available for the Weddell Gyre region between 2002 and 2016 were used in this study to create gridded
fields of velocity and conservative temperature. The latter was derived using the TEOS-10 program in MATLAB (McDougall
et al., 2011). We applied objective mapping to the profile data resulting in a climatology of gridded conservative temperature
on 41 pressure levels between 50 and 2000 dbar. The grid cell resolution varies slightly with changing latitude (Reeve et al.
2019), but is on the order of ~80 x 60 km. This method follows, and is an extension of the objective mapping provided in
Reeve et al. (2016), and the reader is referred to that study for further details on the data quality control and mapping method
used in this study. The absolute velocity field was derived from Argo float trajectory data at the depth of the float drift (800
dbar in the Weddell Gyre). This process required careful quality control assessments and surface drift corrections following
Park et al. (2005), and, given that under-ice profiles have no geo-located position, all such interpolated trajectories were omitted
from the study. Thus, the derived velocity field exhibits considerable bias to summer conditions. The velocities were
objectively mapped to provide a grid of absolute velocity at 800 dbar. This process is detailed in Reeve et al. (2019).

Following Reeve et al. (2019), a stream function was fitted to the velocity field at 800 dbar through the application of a cost
function. By applying a cost function, the resulting stream function provides a best fit for the entire Weddell Gyre, where the
direction of mean flow at the boundaries are assumed to be parallel to the boundaries. The stream function at 800 dbar provides
the reference level for the relative geostrophic velocity, derived from the gridded density field from Reeve et al. (2016), above.
For full details on this methodology, refer to Reeve et al. (2019), which, in addition to Reeve et al. 2016, is the prelude to this
study. In Reeve et al. (2019), a careful error analysis and a detailed comparison of volume transports to available estimates in
the literature justify the method as a reasonable solution to obtaining a large-scale observation-based estimate of the Weddell
Gyre circulation.

There are, however, important improvements between the stream function provided in Reeve et al. (2019), and the stream
function used in this study. The cost function was adapted to allow for a variable coastline, so that the stream function provides
a solution which includes the southern ice shelf edge east of the Prime Meridian (though not the shelf edge currents, such as
the Antarctic Slope Front, which are not resolved in this study as they are beyond the domain of the Argo floats), which is

excluded in Reeve et al. (2019). To better estimate the velocity field along the southern coast of the Weddell Gyre, a few minor adjustments were made to the gridded velocity field prior to fitting a stream function. Firstly, long-term average velocities derived from mooring data were included at the coast near the Prime Meridian and Kapp Norvegia, to better resolve the flow which follows the coastline as it curves southwards towards the Filchner-Ronne ice shelves (the mooring positions are marked in Fig. 1. See Le Piah et al. 2020 for further information about the mooring data). Secondly, the velocity field at Gunnerus Ridge (also marked in Fig. 1) required special treatment. The trajectories of Argo floats show a tight, bathymetrically steered flow around Gunnerus Ridge, which is lost during the objective mapping process due to larger length scales. Also, while the potential vorticity values to either side of the ridge are the same, the direction of the flow is opposing (i.e., primarily northwards on the eastern flank of Gunnerus Ridge and southwards to the west of Gunnerus Ridge; see Fig. 11 in Reeve et al. 2019). This opposing direction is averaged out in the objective mapping. Therefore, after the objective mapping, the closest grid cells to Gunnerus Ridge are replaced with direct velocity measurements derived from the three Argo floats that drift along the ridge (the Argo floats have WMO numbers: 7900164, 7900166 and 7900168). There are caveats to this decision, in that these are data points from three floats with a limited time span (from February-May 2007, and then December 2007 until April 2008), during a period when the area is ice-free. However, by making this adjustment, we improve the performance of the cost function in providing a stream function representative of the large-scale circulation, which includes a more complete inflow (in comparison to Fig. 4 in Reeve et al. 2019). The resulting stream function for the vertically integrated flow from 50-2000 dbar is shown in Fig. 1, where the streamlines curve around Gunnerus Ridge, indicating the main inflow into the southern limb of the Weddell Gyre.

## 3 Methods

### 3.1 The heat budget

Due to conservation, the rate of change of heat storage of a certain ocean volume needs to equal the sum of all fluxes comprising advection and turbulent diffusion, both vertical and horizontal. Thus, the heat budget integrated for a volume of water not in contact with the atmosphere is defined as:

$$\frac{d\Theta}{dt} = -\nabla_H(U \cdot \Theta) - w_E \frac{d\Theta}{dz} + \kappa_H \nabla_H^2 \Theta + \kappa_V \frac{d^2\Theta}{dz^2} + R \tag{1.1}$$

$$\rho_0 C_p \int_{mT-1000}^{mT} \frac{d\Theta}{dt} dz$$
$$= \rho_0 C_p \left\{ \int_{mT-1000}^{mT} (-\nabla_H(U \cdot \Theta) + \kappa_H \nabla_H^2 \Theta) \, dz - w_E(\Theta_{mT} - \Theta_{mT-1000}) + \kappa_V \left( \frac{\partial \Theta}{\partial z}\Big|_{mT} - \frac{\partial \Theta}{\partial z}\Big|_{mT-1000} \right) + R \right\} \tag{1.2}$$

where $\nabla_H$ is the horizontal divergence operator, $U$ is the horizontal geostrophic velocity, $\Theta$ is the conservative temperature, $w_E$ is the vertical velocity (defined as the Ekman pumping velocity; see Supplements S3), z is depth and $\kappa_H$ and $\kappa_V$ are the horizontal and vertical diffusivity respectively (adapted from Tamsitt et al. 2016). R represents the unresolved transient processes excluded in this study (discussed below). For the vertical integration in Eq. 1.2, the subscript $m_T$ describes the mid-point of the thermocline, which provides the upper boundary, while $m_T +1000$ describes the lower boundary. This is to avoid incorporating highly seasonally variable surface waters from the analysis whilst also fixing the volume of water; detailed explanation of the vertical boundaries is provided in the Supplements S1. Figure 2 shows selected vertical profiles with the upper and lower boundaries marked (the corresponding position of the profiles is found in Fig. S1, selected at random to provide a broad coverage of the Weddell Sea). Note the upper boundary temperature is always less than the lower boundary temperature within the Weddell Sea (there are regions where the opposite is true to the north of the gyre, within the ACC). Each term is multiplied by the specific heat capacity of seawater, $C_p$ (~4000 J $K^{-1}$ $Kg^{-1}$), and seawater density, ($\rho_0$ = 1027 kg $m^{-3}$), and integrated for a 1000 m thick layer so that units of each component are given in W $m^{-2}$. The first term on the right-hand side in Eq. 1.1 describes the mean horizontal geostrophic heat advection, where $U$ is derived from horizontal

differentiation of the geostrophic stream function derived from Argo float data (i.e., where $u = \partial\psi/\partial y$ and $v = -\partial\psi/\partial x$; see Section 2.2 and Reeve et al. 2019). Since we derive velocity from a non-divergent stream function, we assume geostrophic flow conditions and omit ageostrophic advection from the first term. This is a pertinent assumption given that the Ekman Layer is excluded from the analysis. The second term on the right-hand side in Eq. 1.1 describes the mean vertical heat advection. The third and fourth terms in Eq. 1.1 (or, the second part of the first term and the third term in Eq. 1.2) describe the horizontal and vertical turbulent heat diffusion components respectively. The sum of these terms results in an estimate of heat tendency over time ($d\theta/dt$), which can be used to determine mean temperature change for a column of water, although this method results in an accumulation of associated errors of the individual terms, and should therefore be treated with caution. Note that given the constraints of the method used (i.e., our data resources are an objectively mapped long-term mean temperature field, and horizontal velocity derived from a long-term mean stream function of the Weddell Gyre, derived from in situ observations), we are unable to look at deviations from the mean, i.e., transient processes. This means, that the meaning of advective and diffusive heat fluxes are different from the ones quantified by Tamsitt et al. (2016). Their underlying numerical model resolves large-scale variations of the flow and temperature field and (partly) mesoscale eddies; these processes are part of the advection, while turbulent diffusion refers to unresolved small-scale processes. In our study, advection is computed from time mean quantities while the effects of mesoscale eddies are parameterised by horizonal turbulent diffusion. Large-scale variations of the flow field are not accounted for in our study. Thus, in addition to the unknown time variability term, $d\Theta/dt$, we also have an unknown 5[th] term in the heat budget in Eq. 1.1 and 1.2, R, which represents the unresolved transient processes excluded from the study. Increased spatial and temporal coverage of in situ observations within the Weddell Gyre would be required to address these gaps (further discussed in section 5.2.2). Maps of the different components are provided in the results section.

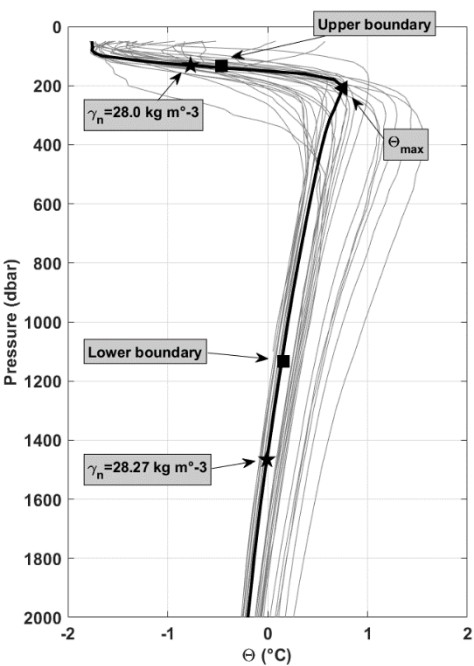

**Figure 2: (a) A sample of vertical profiles of conservative temperature (the positions of the profiles are marked by stars in Fig. S1, the red star marking the position of the example profile in black), where Θmax is marked by a triangle, the upper boundary (i.e., mid-thermocline) and the lower boundary (mid-thermocline + 1000) are marked by squares, and the upper and lower WDW boundaries (defined by a neutral density range of 28 to 28.27 kg m$^{-3}$) are marked by stars. This is to highlight that our method for the vertical boundary limits allows for the full inclusion of the core of WDW while also excluding Winter Water.**

There are two further unknowns in Eq. 1. These are the horizontal and vertical diffusivities, $\kappa_H$ and $\kappa_V$ respectively. For the northern and eastern outskirts of the Weddell Gyre, we define $\kappa_H$ from the dataset provided by Sevellec et al. (2022), who

derive horizontal diffusivities directly from Argo float trajectories by fitting a "pseudo-trajectory" to increase the spatial resolution required for the computation. Given this requires trajectory data without gaps in the record, estimates are missing for much of the Weddell Sea due to the presence of sea-ice. For the remainder of the Weddell Gyre, we define $\kappa_H$ as $247 \pm 63$ $m^2s^{-1}$ and $\kappa_V$ as $(2.39 \pm 2.83) \times 10^{-5}$ $m^2s^{-1}$ based on the estimates provided by Donnelly et al. (2017), which are derived from ship-based observations throughout the Weddell Sea in combination with velocity estimates from the ECCO live access server.

(See Supplements S4 for our reasoning and maps of $\kappa_H$). The implications of defining the diffusivities in this way are discussed in Section 5.

### 3.2   Assessing the uncertainty

The errors for each heat budget term are calculated using the laws of propagation, as detailed in the Supplement to this paper (S7). The main sources of error are from the variables: temperature, horizontal velocity, vertical (Ekman) velocity, and

unknown diffusivity, which is assumed constant in the horizontal for the Weddell Gyre interior and southern limb, and throughout the whole Weddell Sea in the vertical.

We used the objective mapping error to represent the error for temperature (Fig. S7), which is provided in Reeve et al. (2016), and assumed to be the dominating source of error for temperature. This means that the error is also representative of the length scales applied in the objective mapping (Reeve et al., 2016). In Reeve et al. (2016), the length scales were assigned based on

an investigation which showed that 95 % of the grid points have at least 40 Argo profiles within a distance of 500 km (we were then able to reduce this to 400 km in Reeve et al. 2019), which was thus the length scale applied in the second stage of the objective mapping, along with a fractional scale on the effect of f/H, which alters the shape of the area of influence about a grid point from circular, when the bottom bathymetry is flat, to elongated, when a grid point is in close proximity to bathymetric gradients (Reeve et al., 2016). The fractional scale on the effect of f/H allows us to take into account the tendency

of water parcels to follow paths of constant f/H in order to conserve vorticity (for a visual demonstration of the impact of including f/H in the scaling, see Fig. 9 in Reeve et al. 2016). Since the focus is a climatological mean from 2002 to 2016, the applied length scales are chosen for the mapping to represent the large-scale field of the entire Weddell Gyre (Reeve et al., 2016). The resulting mapping errors are large in regions where bathymetry is complex and data coverage is sparse, and low in regions where the bathymetry is flat or where data density is high. Thus, the regions of largest uncertainty include the northern

periphery of the gyre, where data are relatively abundant but the bathymetry is complex, and the eastern edge of the Weddell Gyre, where data are sparse (Fig. S7).

The error for horizontal velocity was provided in Reeve et al. (2019), and is derived from the stream function, where a sensitivity study was implemented, perturbing the velocity field using a combination of factors (mapping errors and drift correction), to provide a range of possible stream function values, from which a standard error was estimated.

Since uncertainty is currently unavailable for the ERA-5 reanalysis data that provided the wind stress field from which Ekman vertical velocity was computed, we took the standard error of the mean of a monthly time series from January 2002 to December 2016 to represent the error for vertical velocity. This represents the natural temporal variability of the vertical velocity, which we assume to be dominating over other possible sources of error, given the large seasonal variability of the wind field.

Lastly, we define an uncertainty range for the diffusivity terms from Donnelly et al. (2017), to be $\pm 63$ $m^2s^{-1}$ and $2.83 \times 10^{-5}$ $m^2s^{-1}$ for the horizontal and vertical diffusivity terms respectively (with the exception of the regions where the Sevellec et al. (2022) dataset is used, where a standard error of the mean is used, see Supplements S4). We discuss this further in Section 5.1.3. The

derivation for the propagated errors is provided in the Supplement (S7), whilst the maps of the error for each heat budget term are provided alongside the heat budget terms in Fig 3. The errors are provided in the large-scale integrations of the IC and SL in Figs. 5-7 and 9 (as pale blocks of colour surrounding the lines).

## 4    Results

Each of the maps provided throughout this paper show streamlines (grey contours) indicating the horizontal circulation of the Weddell Gyre derived from Argo floats (see Section 2.2 and Reeve et al. 2019 for further details). Where the streamlines are closely spaced, flow is more intense than where they are loosely spaced. The streamlines describe a double-gyre structure whereby the western sub-gyre is weaker than the elongated, stronger eastern sub-gyre. While large uncertainties are associated with the eastern part of the eastern sub-gyre, numerical model simulations (e.g., Timmermann et al., 2002), historical observations (e.g., Orsi et al. 1993), and direct volume transport estimates support the concept of a double-gyre structure in the Weddell Sea (e.g., Fig. 7 in Reeve et al., 2019). The following section is presented in two parts. In part one, we provide maps of the vertically integrated heat budget terms (from Eq. 1.2), to obtain an overview of the large-scale heat field of the Weddell Gyre, whereas in part 2, we consider the larger-scale zonal variation of the heat budget.

### 4.1    Part one: the large-scale investigation of heat within the Weddell Gyre

The heat budget contributions from the different terms in Eq. 1.2 (including the signs of each term, e.g., $-\nabla_H U\theta$ and $+\kappa_H \nabla^2 \theta$) are provided in Fig. 3. While for the gyre at large, the mean horizontal geostrophic heat advection (Fig. 3a, left panel) shows a patchwork display of heat transport convergence (positive) and divergence (negative), the southern limb of the gyre is generally dominated by heat transport convergence, of about +20 W m$^{-2}$ west of the Prime Meridian. While small areas of heat flux divergence are found throughout the southern limb, these small areas appear to be less prominent than the areas of positive heat flux and are unlikely to be significant given the high associated errors (Fig. 3a, right panel). The whole region east of the Prime Meridian is dominated by particularly strong patches of positive and negative values in excess of ±80 W m$^{-2}$. Along the northern limb of the gyre, the pattern is dominated by bands of alternating positive and negative values, of about ±60-80 W m$^{-2}$, which are aligned in a manner that appears to follow the complex bathymetry in the region. Note associated errors become increasingly large directly to the north of the northern limb, related to the highly dynamic boundary.

The heat flux due to mean vertical advection ($-w_E(\theta_{mT} - \theta_{mT-1000})$, Fig. 3b, left panel) is positive throughout, and considerably weaker than that due to mean horizontal advection, in the range of ~0-3 W m$^{-2}$. Vertical advection is weakest west of ~10°W, and strongest over the eastern sub-gyre region between 0 and 30° E. That the entire region shows positive vertical fluxes (with the exception of north of the Weddell Gyre) results from two factors: (1) the mean Ekman pumping velocity is positive (indicating upwelling) throughout the offshore Weddell Gyre (downwelling, i.e. negative Ekman pumping velocity, is found in regions shallower than 2000 m and thus outside of our region of Argo float data availability) (Fig. S3), and (2) the upper boundary displays lower temperatures than the lower boundary (Fig. 2), which is a consequence of the vertical boundaries applied in this analysis. Positive vertical advection implies that more heat is advected upwards into the core layer of WDW from below than is leaving by advection through the top, implying that the net effect of vertical advection is to warm the layer in question.

Horizontal turbulent diffusion (Fig. 3c, left panel) is characterised by a positive signal of about 0-40 Wm$^{-2}$ within the gyre interior, and a negative signal along the southern limb in the range of 0-50 Wm$^{-2}$, with the exception of local patches of heat flux convergence such as over Maud Rise and just north of Astrid Ridge at ~10°E (topographic features are marked in Fig. 1). The northern limb of the Weddell Gyre is mostly positive, (20-100 Wm$^{-2}$), though a strip of heat flux divergence sits directly

north of this area of heat flux convergence, in the northern boundary zone between the Weddell Gyre and the Antarctic Circumpolar Current.

As with mean vertical advection, vertical turbulent diffusion (Fig. 3d, left panel) exhibits a uniform sign throughout the Weddell Gyre, with negative values in the range of -1 to -6 Wm$^{-2}$. The strongest values are found in the southern limb, at about ~20°E (~ -6 Wm$^{-2}$), as well as just west of Maud Rise.

The heat tendency resulting from the sum of the heat budget terms (left-hand side of eq. 1.2). is provided in Fig. 3e (left panel, with the corresponding propagation of error provided in the right panel). There is a patchwork of negative (cooling) and positive (warming) values throughout, and overall Fig. 3e is spatially mostly similar to the heat flux due to mean horizontal advection and horizontal turbulent diffusion in Figs. 3a and 3e (the latter especially to the north of the gyre where diffusivity is large). There is, however, a clear warming along the northern limb, driven by horizontal turbulent diffusion (Fig. 3c). Any non-zero value in Fig. 3e should correspond to an area of warming (positive tendency) or cooling (negative tendency) of the water column. Given that data from a 15 years-long observation period have entered the calculation, we wouldn't expect the real ocean to have experienced such a patchy warming and cooling pattern, resulting – in particular – from the horizontal advection field. In section 4.2 we therefore perform spatial integration of the fields displayed in Fig. 3a-e over distinct areas defined by the circulation, in order to eliminate some of the noise, such that more robust statements on the main balances between the different heat flux terms can be made on a regional scale. The uncertainties will be discussed further in section 5.

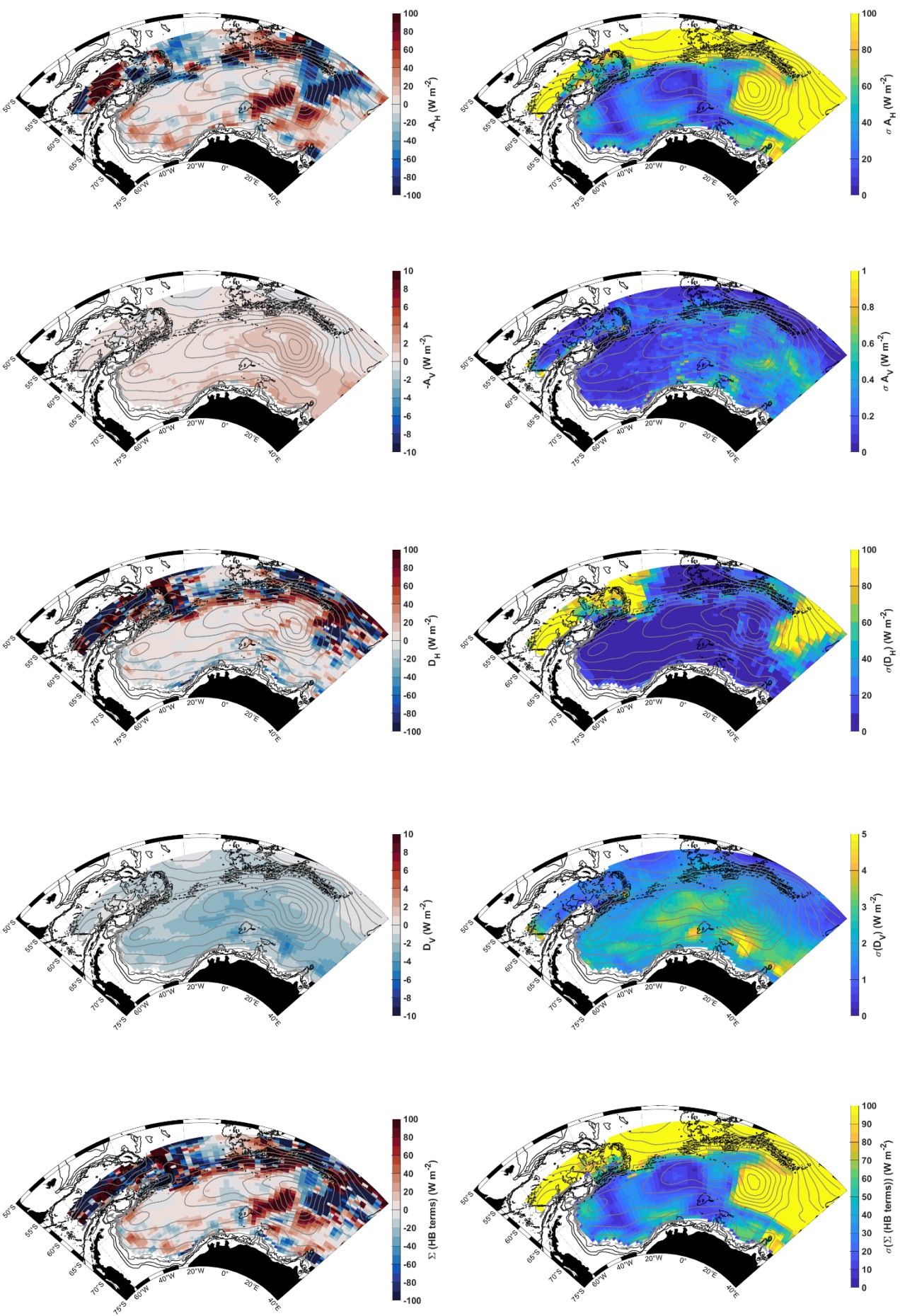

**Figure 3: The panels on the left shows the heat budget terms in W m$^{-2}$, from Eq. 1.2, for a layer of water 1000 m thick from the depth of the mid-thermocline: (a) mean horizontal geostrophic heat advection, (b) mean vertical advection, (c) horizontal turbulent diffusion, (d) vertical turbulent diffusion, and (e) the sum of the terms in a-d. Positive values indicate warming, i.e. heat transport convergence, where more heat is entering the grid cell than is leaving it, whereas negative (blue) values indicate cooling, i.e. heat transport divergence, where more heat is leaving the grid cell than is entering it. The panels on the right show the corresponding total propagated errors of each term. Grey and black contours provide the horizontal streamlines and the 1000, 2000 and 3000 m isobaths respectively (as in Fig. 1). Note the different colour scales for the horizontal versus vertical fluxes.**

## 4.2    Results part 2: zonal variation in the heat budget

In this section, we consider the zonal variation of the heat budget, for two regions: (1) the open southern limb (SL) and (2) the interior circulation cell (IC). The open SL region (i.e., the magenta stippled area in Fig. 4) is defined by the stream function as $16 \leq \Psi \leq 26$ Sv, which describes the open inflow zone where water masses enter the gyre, and spans the entire zonal extent of the double gyre system, from just west of Gunnerus Ridge (~33°E) to ~50° W, where the streamlines veer northwards to follow the coastline of the Antarctic Peninsula. The southern boundary of the SL is the southernmost streamline that does not intersect with the coastline (16 Sv). This definition of the SL enables us to focus on the water that enters the gyre from the east, and circulates the entire zonal extent of the gyre, thus reaching into the south-western interior. The IC region (i.e., the blue stippled area in Fig. 4) is defined as $\Psi \geq 26$ Sv, which is the largest streamline that spans the entire zonal extent of the double gyre system, this time forming a fully enclosed circuit. This definition of the IC allows us to focus on the recirculating waters of the gyre, from just west of Gunnerus Ridge to near the continental shelf edge of the northern tip of the Antarctic Peninsula (~50° W). For both regions, the area east of Maud Rise (3° E) is omitted, due to large uncertainties east of Maud Rise (discussed in Section 5). The analysis is carried out as follows: we compute the mean for each zonal band in our data grids, from just west of Maud Rise (~3° E) in the westward direction towards the Antarctic Peninsula (Figs. 5-6, upper panel). The latitudinal range of the band is defined using the stream function, focusing on the SL and the IC. Lastly, we plot the cumulative zonal integration from east to west (Figs. 5-6, lower panel) and provide the zonally integrated heat budget terms in Table 3. We also take the sum of the heat budget terms and divide by the time period to get the temperature change for SL and IC, also listed in Table 3. For both regions, the zonal variation in the heat budget terms (Fig. 5-6, upper panel) show large local imbalances in the overall heat budget. However, much of the local (grid-scale) imbalances (i.e., the random noise part) cancels out in the net (zonally integrated) heat budget terms, allowing regional patterns not affected by the differentiation at the grid scale to emerge (Fig. 5-6, lower panel).  The resulting volume integration describes the heat flux divergence (negative) or convergence (positive), owing to heat fluxes across the boundaries of the volume of water in question. By considering the 4 different heat budget terms with respect to each other, we can build up a picture of how and where heat is redistributed throughout the Weddell Gyre. Table 2 provides a list of the abbreviations for the terms presented in Figs. 5-10. The method for computing the associated errors is detailed in Section 3.2 as well as the Supplement (S7). We also provide a zonal and meridional analysis of the entire region marked by both blue and magenta stippling in Fig. 4 in the supplements (Figs. S7 and S8). These analyses provide results that agree with the analyses presented in this section and are described in section S8.

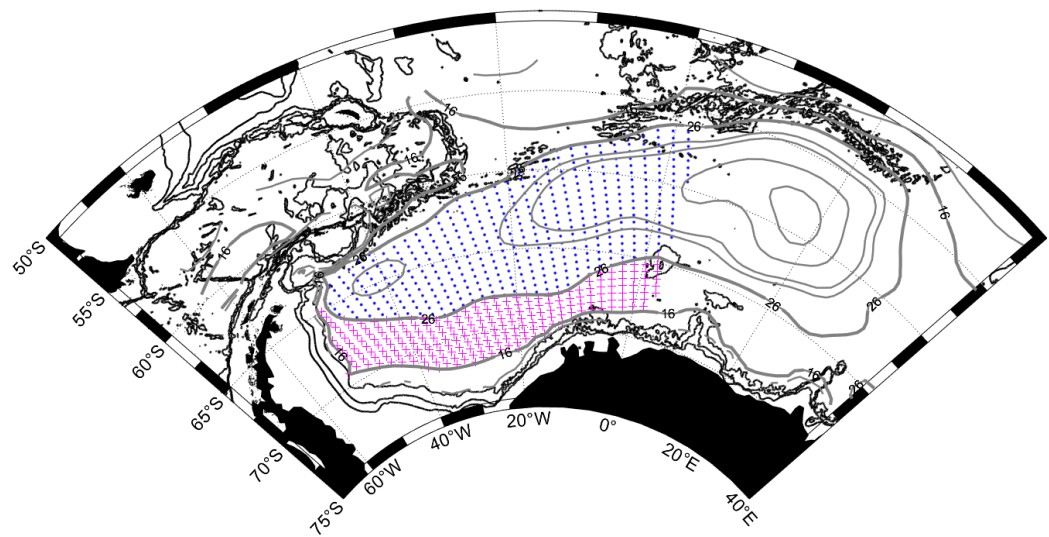

**Figure 4: The stippled areas show the regions defined as SL (magenta crosses) and IC (blue dots) respectively, encased in the streamlines used to define the SL and IC regions (16≤SL≤26 Sv and IC≥26 Sv). These regions are horizontally integrated across in the following Section 4.2. Grey and black contours provide the horizontal streamlines and the 1000, 2000 and 3000 m isobaths respectively (as in Fig. 1).**

### Region 1: SL (southern limb)

Overall, the primary sources of heat (heat flux convergence) are from the advection terms, mean horizontal advection ($A_H$) and mean vertical advection ($A_V$), whereas the primary heat sinks (heat flux divergence) are from the turbulent diffusion terms, horizontal turbulent diffusion ($D_H$) and vertical turbulent diffusion ($D_V$) (Fig. 5, lower panel). The vertical terms are spatially invariant and contribute to the heat budget to a much lesser extent than the horizontal terms. This compliments the findings in Fig. S2 of the net air-sea heat fluxes, which shows a relatively small heat loss through the surface of the ocean in the SL region. The heat flux convergence due to vertical advection is driven by upwelling, which leads to an advection of warm water upwards into the layer, whereas heat flux divergence due to vertical turbulent diffusion removes heat from the layer through the upper boundary (the mid-thermocline). Thus, heat enters the SL primarily through horizontal advection, and to a lesser extent by vertical advection due to upwelling, and is removed from the layer primarily by horizontal turbulent diffusion and vertical turbulent diffusion (a small amount of which may escape through the surface of the ocean). Thus, the turbulent diffusion terms redistribute the heat that is advected into the layer.

The largest regional fluctuations (Fig. 5, upper panel) in the sum of the terms are caused by large fluctuations in mean horizontal advection, with the largest fluctuation occurring downstream of Maud Rise, from about -20 at 0 °E to +20 Wm$^{-2}$ at ~5 °W. There is also an initial removal of ~2 TW of heat from the SL by mean horizontal advection east of 0 °E, possibly related to the influence of Maud rise. Overall, there is a net heat gain due to mean horizontal advection, of ~8 TW, although taking into account the initial heat sink, the amount advected into the layer west of 0 °W is ~10 TW (lower panel of Fig. 5). This influences the net heat tendency (sum of the heat budget terms), which, due to a combination of both the horizontal terms, initially shows a heat sink of ~4 TW between 3 °E and 30 °W, followed by a gradual build-up of heat west of 30 °W, so that the overall net heat flux convergence due to the sum of the heat budget terms for the SL is 0.3 ± 3 TW. The perturbation in mean horizontal advection at Maud Rise is an artefact of being unable to resolve the regional circulation around Maud Rise, whereas the gridded temperature captures the presence of the relatively cold Taylor Cap situated over Maud Rise (clearly

visible in Fig. 1). This creates strong horizontal temperature gradients on the eastern and western flanks of Maud Rise, i.e., in the direction of the dominant flow. This is discussed further in Section 5.2.2.

**Region 2: IC (Interior Circulation cell)**

Overall, the means of the heat budget terms (Fig. 6, upper panel) are similar in the IC to the SL, with the exception of the mean horizontal advection term, which has a larger mean in the SL ($6\pm13$ Wm$^{-2}$) than in the IC ($0.2\pm11$Wm$^{-2}$) (Table 3). Note the differences in the magnitudes of the net heat budget terms are due to the larger area of the IC (Table 3). As with the SL, vertical terms in the IC remain zonally spatially uniform and have much smaller means than the horizontal terms, with heat being vertically advected into the IC domain (via upwelling) and removed from the IC domain by vertical turbulent diffusion. In contrast, the horizontal terms in IC change sign in comparison to the SL. Horizontal turbulent diffusion ($D_H$) is the primary heat source (i.e., heat flux convergence), whereas horizontal mean advection ($A_H$) is the primary heat sink (i.e., heat flux divergence) (Fig. 6, lower panel). Additionally, in contrast to the SL, the surface heat fluxes in this region (Fig. S2) are positive, implying a positive heat flux into the ocean through the surface. This heat does not appear to cross the thermocline (i.e., the upper boundary of our domain), given the convergence of heat due to vertical advection and the positive vertical velocity values in Fig. S3 imply that, in the vertical, heat is entering the IC domain from below (due to upwelling), and vertical turbulent diffusion provides a heat sink (removal of heat), likely upwards through the thermocline, owing to the strong vertical temperature gradients associated with the thermocline. Thus, the heat entering through the ocean surface is likely to be redistributed by other processes (such as surface transports or ice melt).

The largest regional fluctuations within the IC (Fig. 6, upper panel) in the sum of the terms are caused by two peaks in mean horizontal advection, just west of Maud Rise, where mean horizontal advection fluctuates from -7 Wm$^{-2}$ to 20 Wm$^{-2}$ and then to -25 Wm$^{-2}$ over ~400 km (from 3 °E to 0 to ~5 °E).

**Table 2: Explanations of the abbreviations used in Figs. 3-8.**

| Term | Description |
|---|---|
| $A_H$ | Mean horizontal geostrophic heat advection |
| $A_V$ | Mean vertical heat advection |
| $D_H$ | Horizontal turbulent diffusion |
| $D_V$ | Vertical turbulent diffusion |
| $\sum AD$ | The sum of the heat budget terms in Eq. 1, listed above, where A stands for the horizontal and vertical advection terms and D stands for the horizontal and vertical diffusion terms. |

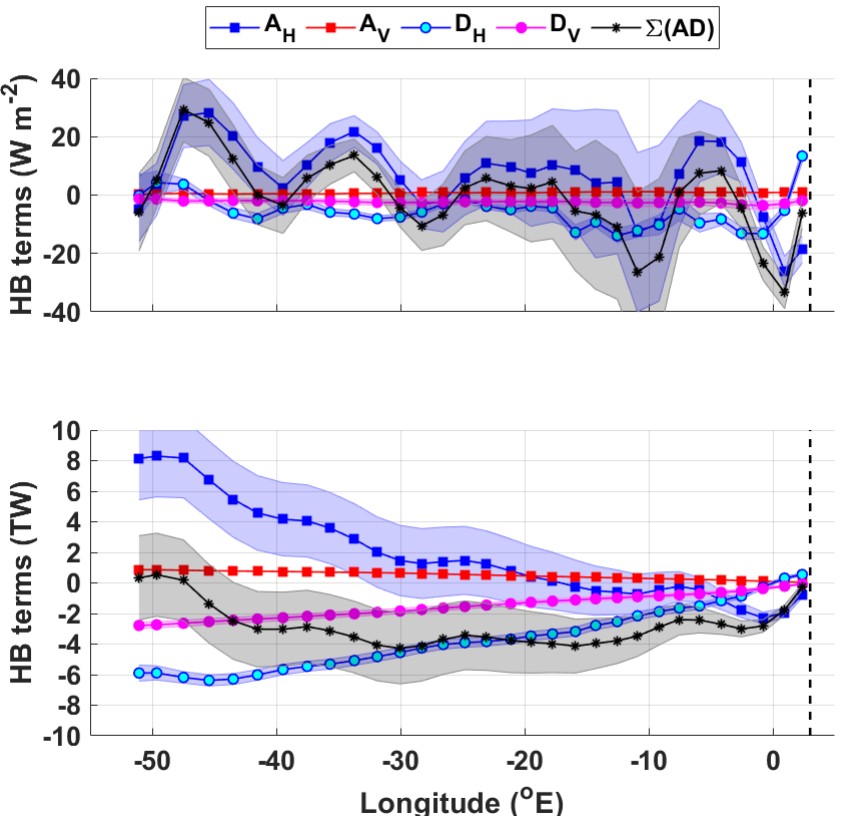

**Figure 5: upper panel: the heat budget terms for the open southern limb (SL) of the gyre in Wm⁻²; lower panel: the cumulative heat budget terms in Terawatts (TW). The key for the legend is listed in Table 2. The dashed vertical line marks the approximate longitude of Maud Rise, at 3º E. The shaded errors provide the associated propagated error (detailed in section 3.2 and the supplement). The SL region is marked by magenta stippling in Fig. 4.**

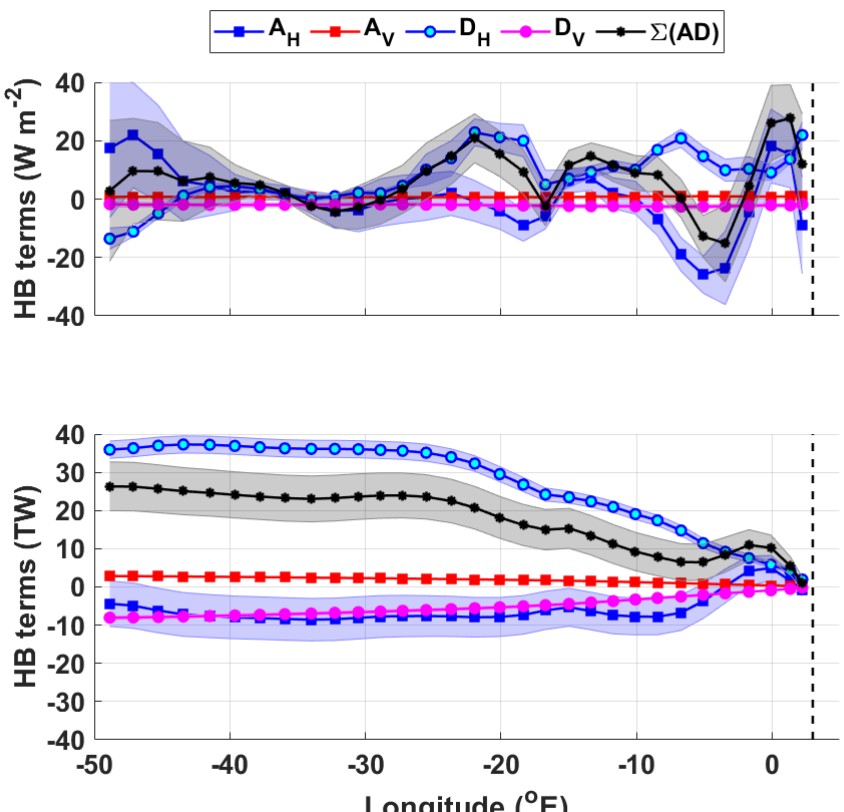

**Figure 6: As in Fig. 5, but for the interior circulation cell (IC) of the Weddell Gyre. The IC region is marked by blue stippling in Fig. 4.**

**Table 3: Zonal mean (W m$^{-2}$) and net (TW) heat budget contribution of the different terms in Eq. 1.2 for the SL and IC zones of the Weddell Gyre (Fig. 5 and 6 respectively), and an estimate of temperature change (dT/dt), using a time period of 14 years. The uncertainty provided for the net heat budget terms (in TW) are the sum in quadrature of the propagated errors. For the mean heat budget terms (in W m$^{-2}$), the provided uncertainty is the standard deviation of the zonal mean heat budget term, and the mean error (in Wm$^{-2}$) is the mean of the propagated error and the standard deviation of the mean propagated error. See Section 5.2 for further information.**

| Heat Budget Term | SL | | | IC | | |
|---|---|---|---|---|---|---|
| | Mean (W m$^{-2}$) | Mean Error (W m$^{-2}$) | Net (TW) | Mean (W m$^{-2}$) | Mean Error (W m$^{-2}$) | Net (TW) |
| Mean Air-Sea flux | -25 | | -3 | +8 | | +27 |
| Mean horizontal advection | +6 ± 13 | 13 ± 7 | +8 ± 3 | +0.2 ± 11 | 9 ± 5 | -4 ± 6 |
| Mean vertical advection | +0.7 ± 0.3 | 0.1 ± 0.04 | +0.9 ± 0.02 | +0.8 ± 0.1 | 0.04 ± 0.01 | +2.8 ± 0.03 |
| Horizontal turbulent diffusion | -5.5 ± 5.7 | 2 ± 1 | -6 ± 0.5 | +7.8 ± 9 | 3 ± 2 | +36 ± 2 |
| Vertical turbulent diffusion | -2.3 ± 0.5 | 1 ± 0.2 | -2.8 ± 0.2 | -2.1 ± 0.3 | 0.6 ± 0.1 | -8 ± 0.4 |
| Heat tendency (i.e., sum of the heat budget terms) | -0.9 ± 14 | 13 ± 7 | +0.3 ± 3 | +6.7 ± 10 | 9 ± 5 | +26 ± 6 |
| Temperature tendency over 14 years (°C) | | | +0.03 ± 0.2 °C or +0.002 ± 0.02 °C/yr | | | +0.8 ± 0.2 °C or 0.06 ± 0.01 °C/yr |

In order to learn more about how the heat budget terms act to redistribute heat throughout the IC, we further split the IC into IC-north (Fig. 7) and IC-south (Fig. 9), where the interface between the two regions is defined as the central gyre axis (i.e., the zonal maximum stream function, north of which the flow is predominantly eastward, south of which the flow is predominantly westward). To further investigate horizontal diffusion, we computed horizontal turbulent diffusion heat fluxes across three zonal boundaries: (1) across the northern boundary of the gyre (defined as $\Psi = 26$ Sv for the northern limb of the gyre, Fig. 4); this is the streamline that, in Fig. 3c, marks the boundary between heat flux convergence within the Weddell Gyre (red) and heat flux divergence to the north (blue)); (2) across the interface between the divergence zone within the SL and the convergence zone in the IC; and (3) across the central gyre axis from IC-south into IC-north. The zonal variation in heat flux for each boundary is provided in Fig. 7b and 9b-c respectively, whereas the zonal integrations of the fluxes are provided in the Supplements (Fig. S6).

Horizontal turbulent diffusion is the dominating heat source to the IC-north, providing a net $34 \pm 2$ TW of heat, whereas horizontal mean advection is a much weaker net heat sink, removing $6 \pm 5$ TW (Fig. 7a, lower panel). Half the horizontal turbulent diffusive flux of heat occurs along the northern boundary of the gyre ($17 \pm 1$ TW; Fig. S6a), with particularly large southward heat fluxes ($>200$ Wm$^{-2}$) directly downstream of the South Sandwich Trench (Fig. 7b), indicating that the rest of this heat flux occurs in the easternmost part of the IC-north (the heat transfer occurring from the IC-south across the central gyre axis into the IC-north is $0.5 \pm 0.4$ TW, and so can be ruled out; Fig. S6c). The large fluxes are most likely due to the strong meridional temperature gradients characteristic of the boundary between the warmer ACC to the north and the colder Weddell Gyre to the south.

Overall, when looking at the zonal variation in the heat budget terms in Fig 7a (upper panel), the positive peaks in horizonal turbulent diffusion are synchronous with the troughs in horizontal mean advection, implying that as heat is turbulently diffused across the boundary into the IC-north, horizontal advection "carries" this heat away along the eastward-flowing northern limb of the gyre. We can also demonstrate this by following the evolution of the sub-surface temperature maximum along a single streamline (in this case, $\Psi=26$ Sv, which is the outer boundary of the IC, as shown in Fig. 4). There is an overall decrease in temperature along the southern limb of the Weddell Gyre and an overall increase in temperature along the northern limb of the Weddell Gyre (Fig. 8).

(a)

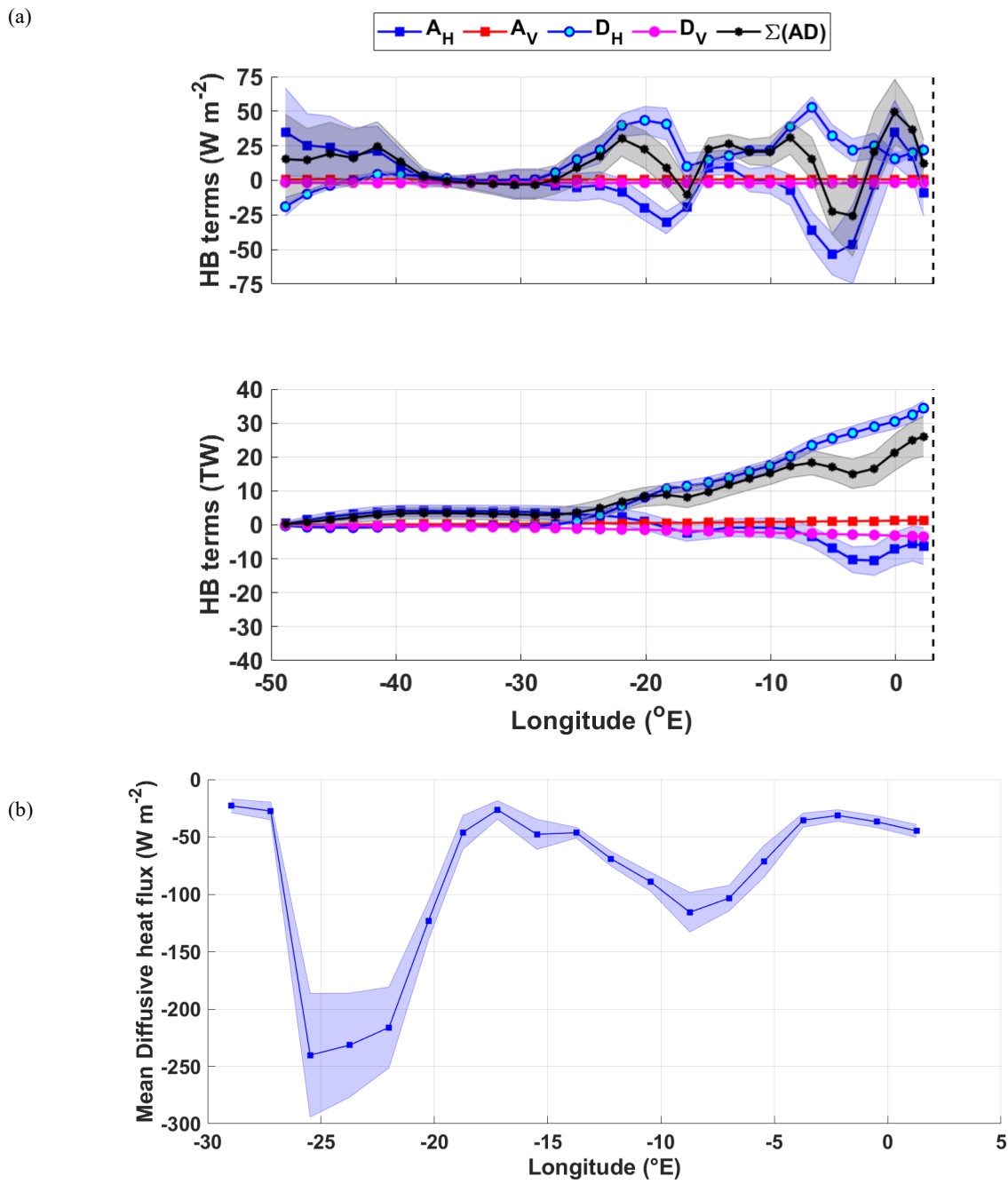

(b)

**Figure 7: (a) Heat budget terms for the IC-north: upper panel: zonal means in Wm⁻²; lower panel: the cumulative heat budget terms from west to east in Terawatts (TW). The key for the legend is listed in Table 2. The dashed vertical line marks the approximate longitude of Maud Rise, at 3° E. Panel b shows the zonal variation of the diffusive horizonal heat flux across the northern boundary of the northern limb of the Weddell Gyre, in W m⁻², defined by the streamline that equals 26 Sv. Negative values indicate a southward flux of heat into the eastward-flowing northern limb of the Weddell Gyre from north of the northern Weddell Gyre boundary (the subsequent cumulative horizontal diffusive heat flux across the northern boundary is provided in the Supplements in Fig. S6a).**

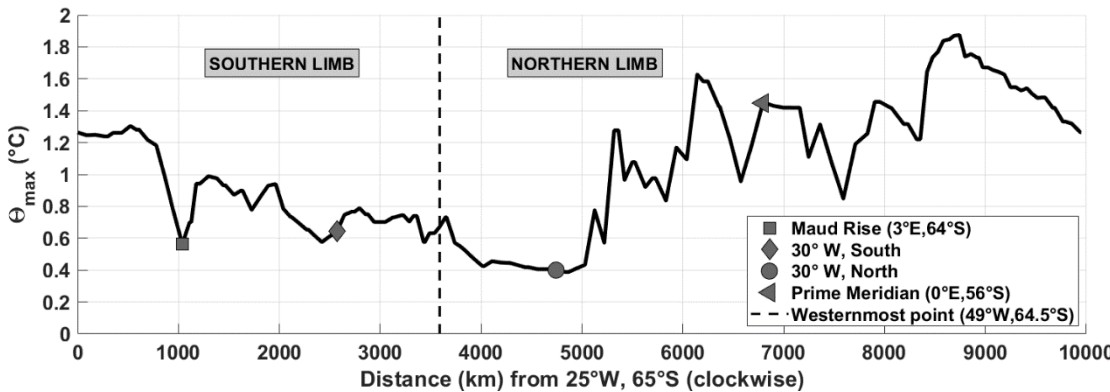

**Figure 8: θ_max (°C) along the streamline Ψ = 26 Sv (i.e., the outermost boundary of the IC). The distance in km is along the streamline in a clockwise direction from 25°W, 65°S, with key locations marked using the legend.**

Horizontal mean advection provides a heat sink of $3 \pm 3$ TW in the IC-South directly west of the Prime Meridian, up until
~10°W (Fig. 9a), after which it provides a heat source of ~5 TW, so that the net heat flux convergence due to mean horizontal
advection is $3.2 \pm 2.8$ TW. The initial heat sink could be indicative of heat advection following the mean circulation about the
eastern sub-gyre, as indicated by the streamlines in Fig. 3; west of this, heat is advected into the area that circulates the western
sub-gyre. In contrast to the SL, horizontal turbulent diffusion is also a source of heat to this area ($1.6 \pm 0.4$ TW), which is
indicative of a northward diffusive heat flux, which removes heat from the SL, transporting it northwards into the IC-south.
This is indeed the case when computing the diffusive heat flux across the northern boundary of the SL in Fig. 9b, which gives
a net heat flux of $1.1 \pm 0.1$ TW (Fig. S6b). Additionally, a net diffusive heat flux of $0.5 \pm 0.04$ TW occurs northwards from
the IC-south across the central gyre axis into the IC-north, mainly occurring 10-28°W, i.e., at the interface between the eastern
and western sub-gyres (Fig. 9c & S6c). The net heat sink in the IC-south is the vertical turbulent diffusion term (Fig. 9a),
which removes $5 \pm 0.3$ TW upwards through the thermocline.

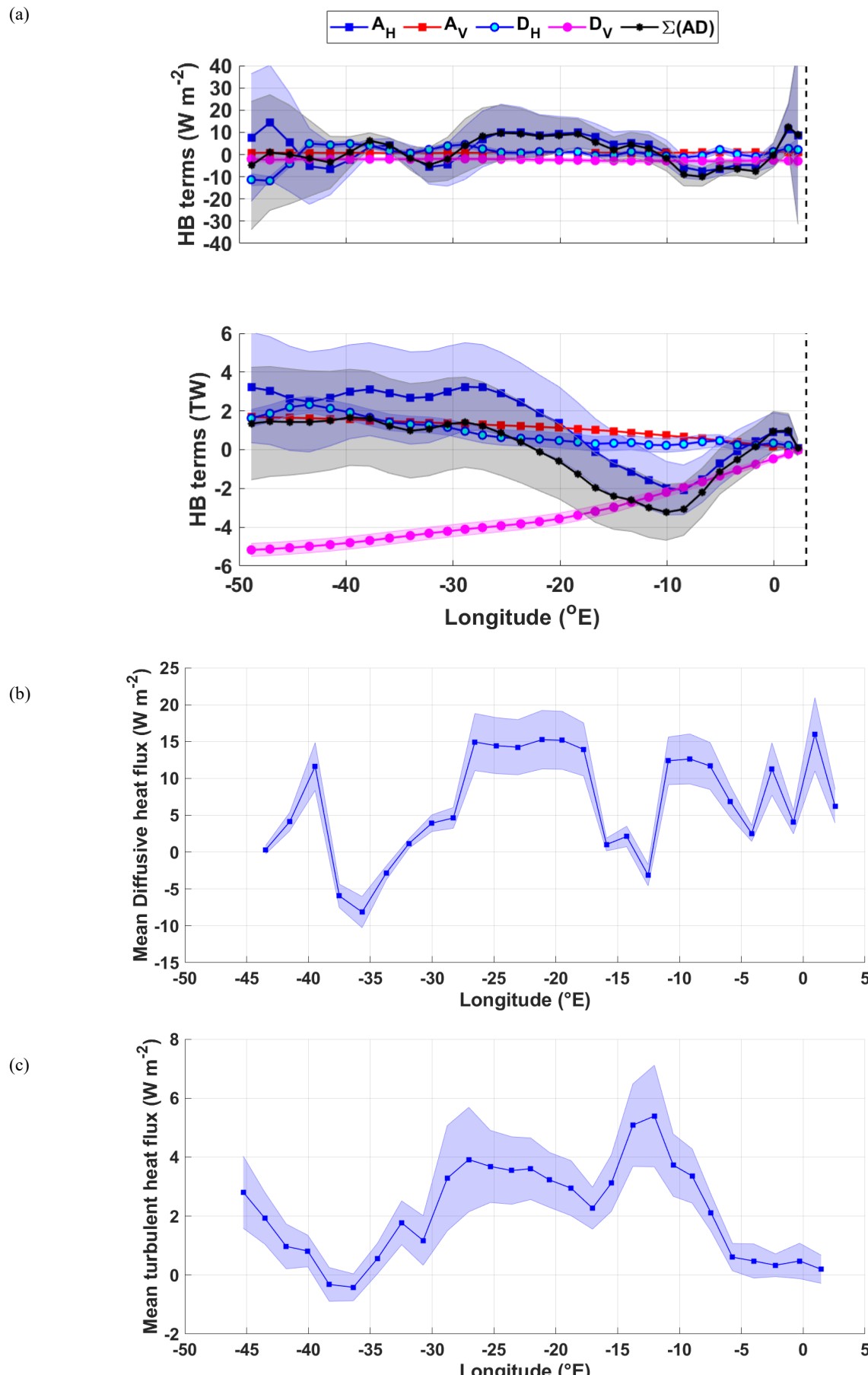

**Figure 9: (a) Heat budget terms for the IC-south: upper panel: the zonal mean heat budget terms in Wm⁻²; lower panel: the cumulative heat budget terms in Terawatts (TW). The key for the legend is listed in Table 2. The dashed vertical line marks the approximate longitude of Maud Rise, at 3° E. Panels b and c show the zonal variation of the diffusive horizontal heat flux in W m⁻² for (b) across the boundary between the SL and the IC, and (c) across the central gyre axis from the IC-south to the IC-north.**

**Positive values indicate a northward flux of heat from the SL into the IC, and from the IC-south northwards into the IC-north, respectively (the subsequent cumulative horizontal diffusive heat flux across the northern boundary is provided in the Supplements in Fig. S6b-c).**

## 5    Discussion

In this study, velocity and temperature derived from Argo floats were used to determine the heat budget of the Weddell Gyre, for a 1000 m thick layer of water extending from the mid-point of the thermocline, encompassing the core of the WDW layer. In the following section, we provide an in-depth discussion of the study limitations, before ultimately discussing and interpreting the results in Section 5.2.

### 5.1    Study limitations

Before interpreting the results, several limitations of the study require discussion.

#### 5.1.1    Vertical boundary limits

The first limitation relates to the omission of the upper 50 dbar from the profiles prior to mapping (Supplements S1), to avoid the highly seasonally varying surface waters. Ideally, we would include the upper ocean layer, in order to explicitly apply the net air-sea heat flux term (Qnet) as a surface boundary condition in the heat budget equation. Since we are interested in the re-distribution of heat throughout the Weddell Gyre (with the main heat source being the WDW), we applied boundary conditions which ensure that the Winter Water layer (defined by its temperature-salinity minimum) is not included within the subsurface layer of interest, while the core of WDW (i.e., $\Theta_{max}$) is fully included. The vertical boundary conditions allow us to consider the varying depths of the upper boundary, while also avoiding bias by fixing the thickness of the vertical layer. However, this, along with the non-uniform resolution of the vertical grid, may introduce some noise into the analysis from grid cell to grid cell, owing to the different depths of the water column the heat budget is integrated over. This may partly explain why there is considerable noise in the maps in Section 4.1, while the noise partially cancels out in Figs. 5 and 6. The implication is that our heat budget analysis is reasonable on large-scales, but would introduce considerable noise when assessed on local-scales. We return to this when discussing regional imbalances in Section 5.2.2, and find that the lateral gradients in the upper boundary depth are unlikely to be a major source of the noise in Fig. 3 (Fig. S5a).

#### 5.1.2    Estimating vertical velocity

Another limitation in the study is a suitable estimate for vertical velocity. Vertical Ekman pumping velocity is used to represent vertical velocity, which is held constant with depth (Supplements S3) throughout the 1000 m thick layer, thus assuming a perfect geostrophic (thus non-divergent) flow. However, reanalysis-based wind products from which $w_E$ is computed, are also limited in their accuracy, largely due to lack of in-situ wind measurements in the Southern Ocean. Additionally, vertical velocities resulting from eddies, fronts and along steeply sloping bathymetry are not taken into account in this study, which might be important sources of vertically non-uniform vertical velocities in the upper ocean.

#### 5.1.3    Diffusivity

The work presented within this paper is based on a long-term mean gridded dataset derived from Argo floats between 2002 and 2016 (Section 2.2). Thus, we are unable to directly incorporate an eddy component into the analysis, which requires fluctuations from the mean temperature and velocity (i.e., u'T'). Unresolved eddying fluctuations are most likely to be an important factor when assessing the heat budget east of the Prime Meridian in the Weddell Gyre, a region we know to be dominated by a mesoscale eddy field (e.g., Schröder & Fahrbach, 1999; Leach et al., 2011; Ryan et al., 2016). We therefore exclude the region east of Maud Rise in the regional analysis in section 4.2. Meanwhile, within this study, we rely on the parameterization of eddy-influences through the estimation of horizontal turbulent diffusion, and the parameterization of vertical instabilities through the estimation of vertical turbulent diffusion. Horizontal and vertical diffusivity ($\kappa_H$ and $\kappa_v$ in Eq.

1) yield considerable uncertainty to this study, especially where they are assumed constant in space. Horizontal diffusivities have been shown to be length scale-dependent (Okubo, 1971; Ledwell et al., 1998). For example, Ledwell et al (1998) demonstrate the importance of length scales in the Eastern North Atlantic by releasing and following the dispersal of a patch of sulphur hexafluoride, estimating values of 2 $m^2$ $s^{-1}$ for length scales of 1-10 km, and as much as 1000 $m^2$ $s^{-1}$ for length scales of 30-300 km. For the entire Southern Ocean, Zika et al. (2009) estimate a value of $300 \pm 150$ $m^2$ $s^{-1}$. Within the Weddell Gyre, Leach et al. (2011) provide estimates of $\kappa_H$ and $\kappa_v$ derived from observations in the Maud Rise region, and estimate $\kappa_H$ to be in the range of 70-140 $m^2$ $s^{-1}$. Leach et al. (2011) consider their estimate for $\kappa_H$ appropriate, albeit on the low side, for mesoscale eddies (although the station spacing of the observations in their analysis is 55 km; not the ~10 km corresponding to the baroclinic Rossby radius in the region). For the Weddell Gyre, Donnelly et al. (2017) provide estimates of $\kappa_H$ derived from observations (WOCE CTD stations from 37 hydrographic sections) on a larger scale than that of individual eddies, where $\kappa_H$ is estimated as $247 \pm 63$ $m^2$ $s^{-1}$. Cole et al. (2015) use salinity anomalies from Argo floats and velocity fluctuations from the ECCO2 product, which is an ocean state estimate that assimilates data (Menemenlis et al., 2005, 2008) to investigate horizontal mixing and the associated length scales. In Fig. 4 of Cole et al. (2015), at about 50 to 60° S in the Atlantic Ocean (the most southerly extent of the authors' analysis), the estimated values of $\kappa_H$ range from 200 to 2000 $m^2$ $s^{-1}$ in the depth range of 0 to 2000 m. In Sevellec et al., (2022), the authors provide a useful dataset of $\kappa_H$ at 1000 m, derived using Argo float trajectories: this is only possibly when a sequence of positions are available, from which the authors derived a "pseudo-trajectory". Thus, most of the seasonally sea-ice covered Weddell Gyre resembles a data gap, with the exception of the Prime Meridian and the north and eastern peripheries of the gyre. These values vary between 0 and 8000 $m^2s^{-1}$; and are close to zero around Maud Rise, and can be 4000-8000 $m^2s^{-1}$ within the eastern and northern gyre periphery. In this study, we are focused on the large-scale, and have a grid of data representative of the long-term mean from 2002-2016. Our grid cell resolution varies slightly with changing latitude (Reeve et al., 2019), but is on the order of ~80 x 60 km, which is nearly double the station spacing of 55 km in Leach et al. (2011). The length-scales applied in the objective mapping are 800 km in stage 1 and 400 km in stage 2, with a skew on the shape of the radius of influence based on the range in potential vorticity (See Section 2.2 in Reeve et al., 2019), and therefore the length scales upon which the analysis falls would actually be considerably greater than in Leach et al. (2011), but with similarities to the length-scale assumed in Cole et al. (2015) of 300 km. We make the decision to use the Sevellec et al. (2022) dataset where numerous data are available, and, for within the Weddell Gyre, we use the values provided by Donnelly et al. (2017), given, these are situated within the area of interest, namely the western Weddell Gyre. A detailed explanation of our method and reasoning regarding the Sevellec et al (2022) dataset is provided in the Supplements (S4).

Referring now to the vertical diffusivities, Leach et al. (2011) provide particularly small estimates for $\kappa_v$, of 3 x $10^{-6}$ $m^2$ $s^{-1}$ for the core of WDW, which they suggest is due to the timing of the survey – having taken place late spring/early summer, when sea-ice had just melted, and the wind had not yet had time to stir up the water column. Other studies provide larger estimates for $\kappa_v$, for various regions throughout the global ocean. Donnelly et al. (2017) estimate $\kappa_v$ as 2.4 $\pm$ 2.8 x $10^{-5}$ $m^2$ $s^{-1}$. Over rough topography, such as 500 m above abyssal sea mounts on the flanks of the mid-Atlantic Ridge in the Brazil basin, Ledwell et al. (2000) provide an estimate for $\kappa_v$ of 3 x $10^{-4}$ $m^2$ $s^{-1}$. Over flat bathymetry of the ocean's abyss, however, Polzin et al. (1997) and Ledwell et al. (1998) estimate $\kappa_v$ to be about 1 x $10^{-5}$ $m^2$ $s^{-1}$. Naveira Garabato et al. (2004a, 2004b, 2007) provide larger numbers for deep water in the Scotia Seas of 3 x $10^{-4}$ to 1 x $10^{-2}$ $m^2$ $s^{-1}$, which they attribute to breaking internal waves. Also within the Southern Ocean, Cisewski et al. (2005, 2008) acquired 7 x $10^{-4}$ $m^2$ $s^{-1}$ in the upper pycnocline of the Antarctic Circumpolar Current at 20° E and Forryan et al. (2013) obtained a value of 1.9 x $10^{-5}$ $m^2$ $s^{-1}$ based on observations in close proximity to a vigorous frontal system between 60 and 80° E, at the northern edge of the Kerguelen Plateau.

Initially, we attempted to estimate $\kappa_v$ through the use of the Richardson Number, following Forryan et al. (2013), defined as the ratio of buoyancy frequency (N) squared to vertical shear squared, and often used to describe the stability of stratified shear flow. For large Richardson Numbers, the resulting diffusivity tends towards a background diffusivity coefficient provided by

the authors. In the Weddell Gyre, the circulation is mostly barotropic, and so the vertical shear is small, leading to large Richardson Numbers, especially when one is focused on the long-term mean. Thus, with this method, the resulting diffusivity tends back towards a background diffusivity parameter. Hence, based on the range of values available from the literature, we made the decision to again use the values provided by Donnelly et al. (2017), of $(2.39 \pm 2.83)$ x$10^{-5}$ m$^2$s$^{-1}$ (the "uncertainty" is incorporated into the error propagation computation).

A final consideration regarding uncertainty is related to using a simple differencing scheme at the grid cell resolution in order to compute spatial gradients in the heat budget equation (Eq. 1.2). Alternatively, we could have extracted curves from the grid from which to determine the gradients. However, this is dependent on further length scales, which could introduce additional uncertainty, and may also result in additional smoothing of the observed fields. Thus, we applied the former method, which extracts the gradients directly from the gridded datasets at the grid cell resolution, with the expectation that this would result in large regional fluctuations owing to the coarse grid resolution.

## 5.2    The Weddell Gyre heat budget

### 5.2.1    Overall findings

While the heat budget does not close on regional scales, it does approximately close when integrating over the open southern limb west of Maud Rise (SL), but not over the interior west of Maud Rise (IC). Nonetheless, important and useful information can be provided from comparing the four resulting heat budget terms.

In most regions of the Weddell Gyre, for the core of WDW, the terms dominating the heat budget are mean horizontal geostrophic advection (Fig. 3a), and horizontal turbulent diffusion (Fig. 3c). Horizontal mean advection appears patchy, though generally implying warming (convergence) in the southern limb of the gyre (Fig. 3a). Horizontal turbulent diffusion (Fig. 3c) acts as a heat source (heat flux convergence) within the eastward flowing northern limb of the gyre and to a lesser extent the gyre interior. In contrast, horizontal turbulent diffusion is associated with a divergent heat flux (or loss of heat) in the open southern limb of the gyre, as well as directly north of the gyre (downstream of the South Sandwich Trench). For the most part, mean vertical advection and vertical turbulent diffusion are spatially relatively uniform, where upwelling advects heat upwards into the WDW layer, and vertical turbulent diffusion removes heat through the top of that layer, upwards through the thermocline to the overlying Winter Water (WW). Additionally, the sum of the heat budget terms in Fig. 3e imply no clear signal in the SL (indeed the heat budget nearly closes when integrating over the region, providing a net gain of 0.3 TW), and a general warming in the IC, synchronous with the surface heat fluxes in Fig. S2, where the IC experiences a net heat flux into the ocean from the atmosphere, while in the SL the ocean loses heat to the atmosphere.

The spatial patterns in the heat budget terms become distinct when integrating over large areas, namely, the SL and IC (Section 4.2). While the vertical terms are spatially uniform throughout, providing a heat source (mean vertical advection) and a heat sink (vertical turbulent diffusion) throughout both regions, the horizontal terms "switch roles" in the SL versus the IC. Horizontal mean advection is a mechanism which brings heat into the SL, while horizontal turbulent diffusion removes much of that heat from the SL. In contrast, horizontal turbulent diffusion brings heat into the IC, whereas mean horizontal advection provides a heat sink in the IC. Thus, while both mean horizontal and vertical advection are responsible for bringing heat into the core of the WDW layer of the Weddell Gyre, turbulent diffusion is the mechanism which then redistributes that heat throughout the gyre interior. Surface heat fluxes (Fig. S2) also imply that the upward diffusive heat flux through the top of the layer (i.e., the mid-thermocline) may represent a source of heat for the observed radiative heat loss in the SL. In the IC, where air-to-sea heat fluxes are positive, the upward diffusive heat flux across the upper boundary of the WDW layer may be horizontally redistributed by the upper ocean flow field and additionally may provide heat required to melt sea ice in this area.

### 5.2.2    Regional (im-)balances

The heat budget shown in section 4 does not close, especially when integrated over smaller areas (Fig. 3e), indicating that noise is largely cancelled when integrating over larger areas. The noise is likely a result of (1) discrepancies in the depth range from grid cell to grid cell, (2) the nature of differentiating across grid cells, (3) due the presence of mesoscale eddies unresolved by the data and methods used, and (4) aliased observations that result in a distortion of the mean state. Here, we discuss the influence of (1) unresolved mesoscale eddies and narrow currents in the eastern Weddell Gyre, (2) Maud Rise and (3) the open northern boundary, and discuss what might be missing that prevents the closure of our heat budget analysis.

*Unresolved mesoscale eddies and narrow currents* - The presence of unresolved mesoscale eddies is particularly important east of the Prime Meridian. In Fig. 1, it is clear there is a misalignment between the streamlines and temperature distribution in the eastern part of the gyre; horizontal mean advection hugely dominates this region which is partially compensated by horizontal turbulent diffusion (Fig. 3). Given the domed shape of the isopycnals characteristic of a cyclonic gyre (e.g., Strass et al., 2020; Fahrbach et al., 2011), we hypothesised that a larger bias due to the horizontal gradient of the upper boundary depth (i.e., mid-thermocline, Fig. S1) was occurring at the gyre periphery (i.e., where the slopes of the isopycnals are largest), which may be contributing to the large positive and negative values that extend diagonally outwards from the centre of the eastern sub-gyre (i.e. forming roughly shaped ellipses) in horizontal mean advection in Fig. 3a. This appears not to be the case, given there is no clear large horizontal gradient in the mid-thermocline depth in the east; indeed, the largest horizontal gradient in mid-thermocline depth occurs in the very west of the Weddell Gyre, along the Antarctic Peninsula, and to a lesser extent over Astrid Ridge (Fig. S5a). The large ellipses do, however, appear to be related to large horizontal temperature gradients (Fig. S5b). We found the largest horizontal temperature gradients unsurprisingly along the northern boundary of the Weddell Gyre (owing to the considerably warmer ACC to the north), but also in a diagonal line spanning from directly east of Maud Rise, northwards to the northeast of the eastern sub-gyre (~30°E, 55°S); in other words, synchronous with the north-easternmost ellipse of heat flux divergence and south-westernmost ellipse of heat flux convergence (Fig. 3a & 3e). The strong horizontal temperature gradients therefore likely accounts for the two strongest ellipses in Fig. 3. This, along with the fact that the computation (which relies on differencing across grid cells) is sensitive to the alignment of the temperature field to the velocity field, which is imperfect, especially on such a coarse resolution grid. In particular, temperature is a more conservative (slow changing) variable in comparison to the velocity field. Thus, in capturing the mean state, our depiction of the mean velocity field may be especially distorted due to aliasing of observations, which we know are biased to summer conditions, and are particularly varying on shorter timescales in the dynamic region east of the Prime Meridian. The summertime bias and coarse resolution likely results in an underestimate of the mean gyre strength, and thus the velocity field derived from the stream function (Neme et al., 2021). In addition to the aliasing of the observations in representing the mean state, it is likely that there is a significant additional component (in addition to mean horizontal advection), such as transient eddying processes, which distributes heat, particularly in the eastern part of the gyre. The question is, how much of this is a result of unresolved mesoscale activity driving large temperature and velocity gradients on much shorter length scales than can be appreciated here, resulting in thus aliased observations, and whether the additional component is a real measurable yet unobserved (in this study) component, or an artefact of the method and data sources used in this study. This is a key consideration, which means that the sum of the 4 heat budget terms, especially in the east, cannot be viewed as the heat tendency (which should be zero in a closed system), but that it additionally consists of unresolved processes (i.e., "$R$" in Eq. 1.1 & 1.2). These include mesoscale eddy activity that is not fully represented by parameterization via turbulent diffusion. We know that the eastern part of the eastern sub-gyre is dominated by an intense mesoscale eddy field (Ryan et al., 2016, Leach et al., 2011 and Gordon and Huber 1984). Wilson et al. (2022) show, using idealised models, that transient eddies are responsible for most of the southward heat transport in the eastern limb of the Weddell Gyre. In addition, as discussed in section 3, a process not accounted for is regional-scale variations in the temperature and flow fields. This process might be particularly important in the eastern sub-gyre, where the boundary to the gyre is poorly-defined due to the openness of the topography. Indeed, Schröder and Fahrbach (1999) suggest

that there is no continuous current marking the eastern boundary, and that baroclinic shear instabilities lead to a breakdown of the eastward-flowing current in the northern limb of the gyre, and that the current "reforms" in the westward-flowing southern limb. The geometry of the eastern sub-gyre might therefore be sensitive to interannual to decadal variations in the wind forcing, potentially affecting the time mean heat flux convergence in this area. These findings in Schröder and Fahrbach (1999) also aligns with the findings of Sonnewald et al. (2023), who use machine learning in a climate model to diagnose the dominating dynamic regimes in the Southern Ocean, leading them to propose a circumpolar "super-gyre" which connects the Weddell and Ross sub-gyre systems. In the eastern sub-gyre region, recirculated "cold-regime" WDW (modified primarily through heat loss) comes into contact with incoming "warm-regime" WDW (Gordon and Huber, 1984). The "warm-regime" WDW represents relatively warm WDW advected into the gyre at the eastern inflow zone at about 30° E, driven by mesoscale eddies (Deacon 1979; Orsi et al. 1993; Orsi et al. 1995; Gouretski and Danilov 1993, 1994; Ryan et al., 2016). When comparing the two terms in Fig. 3a and 3c, while the magnitude is much larger in Fig. 3a (horizontal mean advection), horizontal turbulent diffusion displays the opposite signs and partially compensates in the eastern sub-gyre region in Fig. 3c. The terms do not cancel, and thus imply that the missing "*R*" term is significant in this region (although, large errors associated with mean horizontal advection also imply that mean advection is poorly represented in this region). This is not easily remedied, since the eastern Weddell Gyre is a region with poor data coverage, including from Argo floats, though at the time of writing, efforts are underway to close this key observational gap.

We'd like to acknowledge that our framework of inferring the heat budget is rather traditional, in which we parameterize the effects of eddies by means of horizontal diffusion. A more advanced approach is represented by the temporal-residual-mean framework (McDougall and McIntosh, 2001), in which the effects of eddies are decomposed into eddy-induced advection (adding an eddy-induced velocity to the time mean velocity in the advection term of the tracer equation) and eddy-induced diffusion. The latter can be decomposed into isopycnal and diapycnal diffusion (Groeskamp et al., 2016). We acknowledge this framework to be more physics-based than our classical approach, yet, given the limitation of our dataset, the estimation of the eddy-induced velocities is not straightforward. At the same time, Sevellec et al. (2019) demonstrated the usefulness of the temporal-residual-mean framework when applied to interpreting eddy-driven horizontal buoyancy transports from mooring-based observations acquired in Drake Passage. In particular, they highlight importance of eddy-driven horizontal transports in the direction perpendicular to the mean flow. For future work it would therefore be intriguing to demonstrate, whether the application of this framework to our data set would represent a major step forward towards closing the heat budget in the eddy-rich eastern part of the Weddell Gyre and around Maud Rise, where our approach does not lead to satisfactory results.

*Entrainment* - It could also be that there is a missing process, such as entrainment, that may account for the non-closure of the heat budget terms in Fig. 3e (though entrainment would constitute a heat flux divergence which may help close the heat budget of the IC, but not of the SL). Indeed, Schlosser et al. (1987) used Helium-3 tracers within the north-western Weddell Sea to estimate a vertical diffusivity that is twice the value that we use here ($5 \times 10^{-5}$ m$^2$ s$^{-1}$) along with a rate of entrainment of WDW into the overlying WW of 15-35 m yr$^{-1}$. Behrendt et al. (2011) argue that increasing WW salinity during 1992-1996 is caused primarily by entrainment of WDW, and lists entrainment as one of three dominating causes of salinity changes to WW (the others being sea-ice formation related salt release and horizontal advection). Brown et al. (2015) also highlight the important role of entrainment in the carbon dynamics of the Weddell Gyre, by bringing dissolved inorganic carbon and salt upwards into the WW from the underlying WDW. Given this work is based entirely on observations, and that we lack sufficient observations to effectively resolve the complex dynamics of the eastern Weddell Gyre, the findings here illustrate the need for better observational coverage of the Southern Ocean in the high latitudes east of the Prime Meridian, a region thus far often omitted from observation campaigns.

*Maud Rise* - Maud Rise is a prominent feature in temperature (Fig. 1), mean horizontal advection (Fig. 3a), horizontal turbulent diffusion (Fig. 3c) and to a lesser extent, vertical diffusion (Fig. 3d). The effect of Maud Rise on WDW temperatures is due to

the presence of a Taylor cap directly over Maud Rise, which has been previously observed as a column of relatively cold water surrounded by a warm halo on the flanks of Maud Rise (e.g., Muench et al., 2001; Leach et al., 2011). Regarding the heat budget terms shown in Fig. 3, mean horizontal advection results in a heat flux convergence upstream of Maud Rise, which is partially balanced by heat flux divergence due to horizontal turbulent diffusion. In contrast, downstream of Maud Rise, heat flux divergence occurs due to both mean horizontal advection and horizontal turbulent diffusion in Fig. 3 (although a convergence peak in horizontal turbulent diffusion does occur directly over Maud Rise).

The effect of mean horizontal heat advection on the flanks of Maud Rise is probably mainly an artefact of the velocity field. Since the water overlying Maud Rise is cold, and the velocity field does not adequately resolve the flow circulating the seamount (instead the streamlines in Fig. 1 cut directly East-West across Maud Rise), the heat convergence upstream of Maud Rise and divergence downstream of Maud Rise are caused by strong lateral temperature gradients between the cold water column overlying Maud Rise and the warm halo surrounding it. However, horizontal turbulent diffusion, which responds in a similar manner (since it is determined exclusively from observed horizontal temperature gradients, but not the velocity field, and therefore the response to Maud Rise is much smaller), acts to balance the influence of mean horizontal advection upstream of Maud Rise (Fig. 3c). Indeed, the overall heat loss that occurs downstream of Maud Rise in Fig. 3a is about 30 W m$^{-2}$, which is similar to previous estimates in Muench et al. (2001) and also in McPhee et al. (1999; 52 W m$^{-2}$ west of Maud Rise and 23 W m$^{-2}$ over Maud Rise), although these authors focus on the surface heat flux, both using surface drifting buoys.

Ultimately, we show that heat flux divergence due to horizontal turbulent diffusion occurs on the flanks of Maud Rise, in agreement with previous studies, which show for instance that baroclinic instabilities on the flanks of Maud Rise are the source of recurrent eddies (Akitomo et al., 2006). Furthermore, Leach et al. (2011) and Ryan et al. (2016) suggest a mixing of WDW with modified recirculating WDW downstream of Maud Rise which would explain the initial heat sink between Maud Rise and the Prime Meridian in Fig. 5a.

***The open northern boundary*** - Regarding the northern limb of the Weddell Gyre, our findings imply that horizontal turbulent diffusion is a mechanism by which heat enters the region (Fig. 3c). This is in some agreement with Jullion et al. (2014), who use an inverse model based on ship-based sections along 30º E to the coast and also along the northern periphery of the gyre at about 55-60º S in order to diagnose the heat budget of the full water column. They suggest that most of the heat advected into the Weddell Gyre occurs along the northern gyre periphery, rather than from the eastern periphery, and reaches the southwestern Weddell Gyre through recirculation in the interior Weddell Gyre, leading to an entrainment of heat into the Antarctic Slope Front. This analysis is not able to resolve localised features such as the Antarctic Slope Front, but there is an indication, especially from the streamlines, that recirculation of the eastern sub-gyre plays a role in the distribution of heat in the Weddell Gyre (Fig. 1). Furthermore, in Jullion et al. (2014), eddy-induced transport contributes significantly to the heat budget of the Weddell Gyre, with a heat flux of 5 ± 1 TW, out of a net heat flux of 36 ± 13 TW from the ACC into the Weddell gyre, which is primarily due to mean circulation. The results provided in Figs. 3c, 7 & 9 show agreement with this finding; the Weddell Gyre's northern limb and the IC are dominated by a convergence of heat due to horizontal turbulent diffusion. Both the southern limb and the area directly north of the Weddell Gyre in the ACC, as well as east of ~30°E, in contrast, exhibit heat flux divergence (cooling) due to horizontal turbulent diffusion (Fig. 3c). This suggests that horizontal turbulent diffusion constitutes an important role in transporting heat into the Weddell Gyre along the open northern boundary as well as from the East- (although large uncertainty in the east requires some caution), in agreement with Jullion et al. (2014). The heat fluxes provided here are somewhat more than the eddy induced heat transport from Jullion et al. (2014) across both the northern and eastern boundary, which, however, accounts for the whole water column, across 2 hydrographic sections which represent the entire open boundary of the Weddell Gyre. We expect their eddy heat flux estimate to exhibit major uncertainties, as it is firstly based on a station spacing of the temperature and velocity profiles that are not nearly eddy resolving (especially along the northern boundary of the gyre) and secondly represents a one-time snapshot.

According to Tamsitt et al. (2016) and Naveira Garabato et al. (2011), major topographic features result in a divergence of horizontal and vertical eddy heat fluxes, leading to substantial warming in association with regions of enhanced mesoscale energy. Thompson & Salleé (2012) use particle advection experiments to show that the enhancement of eddy kinetic energy occurs downstream off topographic obstacles, which may explain cross-front exchange associated with jets in the lee of topographic features. This may explain the heat flux divergence due to horizontal turbulent diffusion that occurs in the lee of Maud Rise (Fig. 3c). It may also help to explain the fluctuations along the northern limb of the Weddell Gyre in Figs. 3a and 3e, where the topography is complex, creating an open-ocean northern boundary to the Weddell Gyre. Indeed, the alternating bands of convergence and divergence along the northern limb of the gyre between 30° W and 20° E in Figs. 3a and 3e appear to reflect the underlying bathymetry, though the alternating bands are also likely due to the effects of meandering of the northern boundary on a coarse resolution grid.

*Turbulent diffusion* - There is also a relatively strong heat flux divergence due to horizontal turbulent diffusion along the southern boundary of the Weddell Gyre towards the coastline in Fig. 3c, especially between 40 and 10° W, and between Astrid ridge and Gunnerus ridge (between 10 and 30° E), indicating that horizontal turbulent diffusion may also constitute an important role in transporting heat towards the shelves along the southern coastline. Indeed, enhanced diffusive mixing over the continental slope (region not covered by this study) has been observed in the southwestern Weddell Sea (Fer er al., 2016; Daae et al., 2009). We are unable to directly compute turbulent heat fluxes across the southern boundary of the southern inflow limb toward the shelf edge due to the requirement of differencing across grid cells, and caution should be made with any attempt at inferring fluxes due to the large uncertainty of computing the heat budget at the boundary. Yet, we may estimate the shelf-ward heat flux indirectly. Fig. S6b reveals the net diffusive heat flux across the northern boundary of the southern inflow limb to amount to $1.1 \pm 0.1$ TW. If we subtract this value from the total heat flux divergence due to horizontal turbulent diffusion of the SL of $-6 \pm 0.5$ TW (Table 3), then we can cautiously infer that most of the remaining horizontal turbulent heat flux occurs southwards towards the ice-covered shelf seas west of Maud Rise (Fig. 3c), of $5 \pm 1$ TW (southwards having a much larger length with larger negative values in Fig. 3c in comparison to the western end of the SL region, (i.e., the magenta stippled area in Fig. 4).

These results indicate that horizontal turbulent diffusion may play an important role in transporting heat southwards across the open northern boundary of the Weddell Gyre (Fig. 7b & S6a), and also southwards towards the continental shelves along the Antarctic coast (Fig. 3c). Furthermore, the horizontal turbulent diffusion of heat may allow for the removal of some heat from the southern limb of the Weddell Gyre (Fig. 9b & S6b), before it is able to advect westwards towards the southwestern corner of the gyre, where the fragile large Filchner-Ronne ice shelf and the ice shelves of the Antarctic Peninsula are located (Hellmer et al., 2012), and where recent advance in the understanding of ocean heat fluxes have been made (Ryan et al., 2020). Since the turbulent diffusive heat fluxes are dependent on horizontal temperature gradients (related to geostrophic shear), this implies a complex interaction between the strength of the Weddell Gyre, thus mean horizontal advection, and the rate of meridional turbulent diffusion. Potentially, up to a certain point, meridional turbulent diffusion may provide a buffer, protecting the southwestern gyre from increased advective heat fluxes resulting from an intensified Weddell Gyre, by also increasing in intensity (due to stronger lateral temperature gradients and velocity shear). This mechanism requires careful understanding if we are to understand the role of the Weddell Gyre in the redistribution of heat in a changing climate.

## 6    Conclusions

Gridded climatologies of temperature and velocity derived from Argo floats spanning 2002-2016 were used to determine the heat budget of a 1000 m thick layer encompassing the core of WDW within the Weddell Gyre. This investigation was to establish the mechanisms by which heat is distributed throughout the Weddell Gyre, implicitly assuming non-divergent,

geostrophic flow conditions. The mechanisms are summarised in the form of a basic schematic in Fig. 10, and interpreted below.

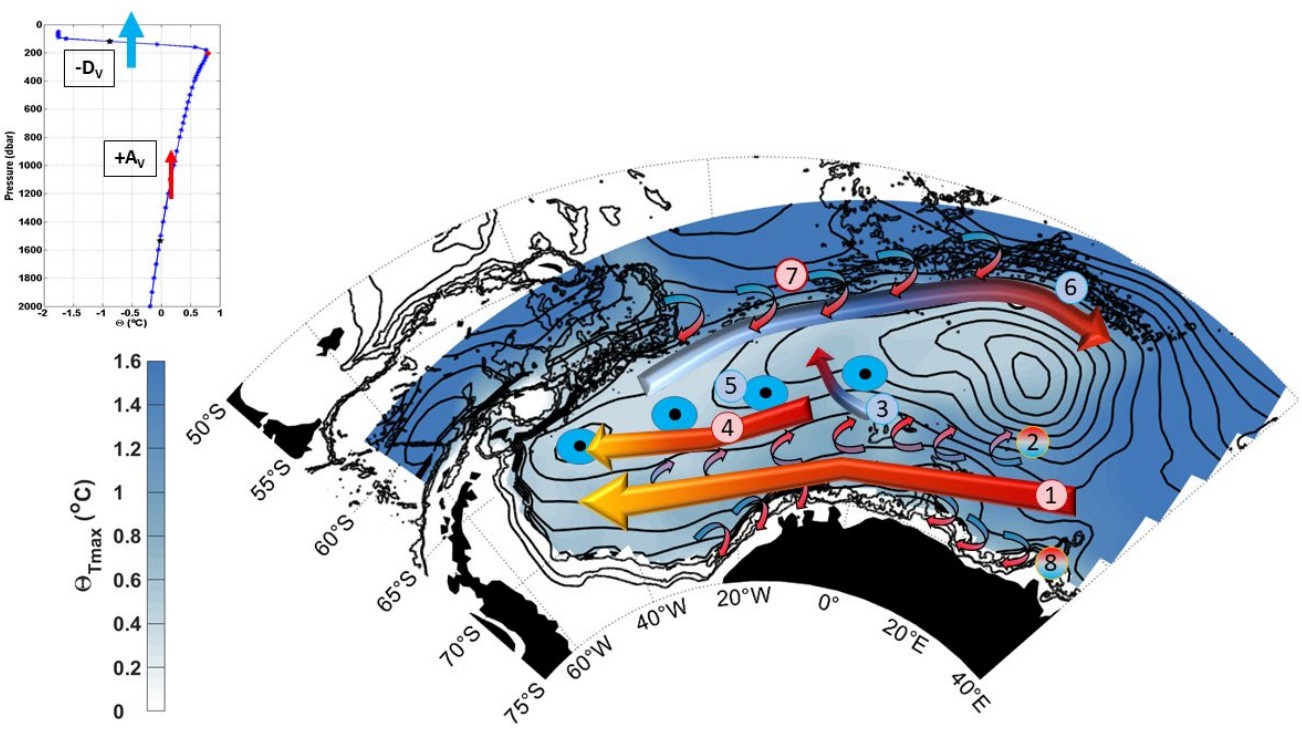

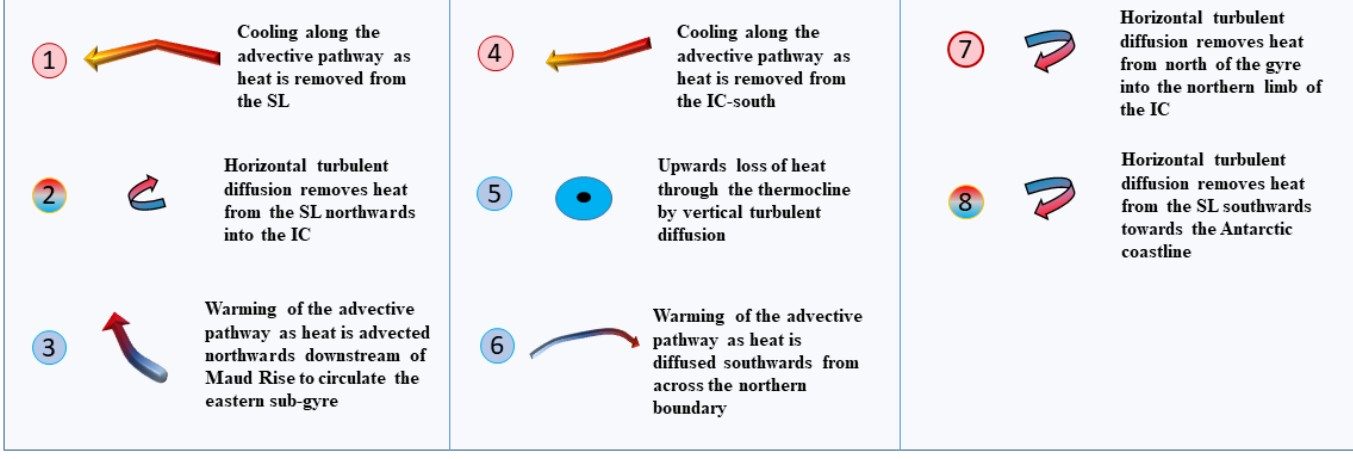

**Figure 10: Schematic of proposed mechanisms by which heat is transported throughout the Weddell Gyre, based on interpretation of results in Sections 4 & 5.2. The blue-scale colour shading shows $\Theta_{max}$ and the black contours show the stream function with 5 Sv spacing, as in Fig. 1. The numbered keys assigned to each feature (arrows and circles) are described in panel b and Table 4. Where the colour of the number and feature are red, heat flux convergence (heat source) is indicated, and where they are blue indicates heat flux divergence (heat sink). The horizontal turbulent diffusive fluxes, indicated by small curved arrows (2,7,8), change from blue to red to indicate a direction of heat flux. The upper left inset shows an example of a vertical temperature profile, with –Dv indicating a vertical diffusion of heat upwards out of the layer through the thermocline, and +Av indicating a vertical advection of heat upwards into the layer from below.**

**Table 4: Summary of the key heat budget terms that are shown in the schematic in Fig. 8. The numbers correspond to the number key in Fig. 10, and the associated features in the schematic. Abbreviations used for the heat budget terms are defined in Table 2.**

| Key | Process | Net heat budget contribution (TW) |
|---|---|---|

| | | |
|---|---|---|
| 1 | A$_H$ into SL | +8 ± 3 TW |
| 2 | D$_H$ out of SL (into IC-south) | -6 ± 0.5 TW (into IC-south: ~1.1 ± 0.1 TW, with a net heat flux convergence in IC-south of 1.6 ± 0.4 TW) |
| 3 | A$_H$ out of IC-south (west of MR and east of 10°W; i.e., circulating eastern sub-gyre) | -2 ± 1.4 TW |
| 4 | A$_H$ into IC-south (west of 10 °W) | 5 ± 3 TW |
| 5 | D$_V$ out of IC-south | -5 ± 0.3 TW |
| 6 | A$_H$ out of IC-north | -6 ± 5 TW |
| 7 | D$_H$ into IC-north from ACC | 34 ± 2 TW (direct flux across northern boundary is ~17 ± 1 TW, suggesting over half exits the eastern end of the IC-north layer) |

If we accept the potential sources of uncertainty discussed in Section 5, we can interpret the results presented in this study as follows:

1. In the SL of the Weddell Gyre, heat *convergence* due to mean horizontal and vertical advection is balanced by a *divergence* of heat due to horizontal and vertical turbulent diffusion. 8 ± 3 TW of heat is advected into the southern limb of the Weddell Gyre (Fig. 10, no. 1), while 6 ± 0.5 TW of heat is removed from the same region by horizontal turbulent diffusion (Fig. 10, no. 2). 1.1 ± 0.1 TW of heat is directly removed from the southern limb by horizontal turbulent diffusion northwards into the IC, which accounts for most of the net heat flux convergence due to horizontal turbulent diffusion in the IC-south (1.6 ± 0.4 TW).

2. In contrast, in the IC, heat *convergence* due to horizontal turbulent diffusion is not balanced by heat *divergence* due to mean horizontal advection. 4 ± 6 TW of heat is advected out of the IC (Fig. 10, no. 6), while 36±2 TW of heat is turbulently diffused into the IC (Fig. 10, no. 7).

3. When we consider the eastward flowing northern limb and the westward flowing southern limb as separate regions, we see that the heat flux *convergence* due to horizontal diffusion mostly occurs in the northern limb of the IC. Half the net heat flux *convergence* into the IC-north of 34 ± 2 TW is due to a southward heat flux across the northern boundary of the gyre (17 ± 1 TW, Fig. 10, no. 7), the remainder mostly from the eastern gyre. A net heat flux *convergence* due to horizontal turbulent diffusion also occurs in the IC-south of 1.1 ± 0.1 TW (Fig. 10, no. 2). Horizontal mean advection acts to remove heat from the northern limb of the IC (-6 ± 5 TW; Fig. 10, no. 6).

4. There is an advective pathway of heat northwards from the gyre's southern limb between 10 and 20° W, which has an impact on the spatial distribution of $\Theta_{max}$, and removes 2 ± 1.4 TW from the IC-south (i.e., north of Maud Rise), between 0°E and 10° W to circulate the eastern sub-gyre (Fig. 10, no. 3). West of 10° W, heat is advected into the IC-south (5 ± 3 TW) (Fig. 10, no. 4). The main mechanism that removes heat from the IC-south is vertical turbulent diffusion (heat is lost upwards through the thermocline with a net flux of -5 ± 0.3 TW; Fig. 10, no. 5).

5. At the southern boundary, there appears to be a diffusive flux of heat towards the shelf seas along the southern coastline, especially between 10 and 30° W, and 10 and 30° E. This may account for the remaining heat flux divergence due to horizontal turbulent diffusion along the southern limb of the Weddell Gyre (i.e., -6 ± 0.5 TW in table 3, of which 1.1 ± 0.1 TW is diffused northwards (Fig. 10 and Fig. S6b), leaving a divergence of 5 ± 2 TW to be

diffused southwards along the southern boundary in Fig. 10, no. 8). While errors are high at the boundary since Argo floats cannot fully resolve shelf edge boundary currents, this may indicate that horizontal turbulent diffusion plays an important role in delivering heat towards the ice-covered shelf seas.

6. In contrast to the horizontal terms, the vertical terms are spatially uniform, where the vertical advection term leads to heat flux convergence throughout both the SL and IC while the vertical turbulent diffusion term results in heat flux divergence. The terms are nearly negligible in comparison to the horizontal terms. Given that Ekman pumping velocity is mostly positive upwards throughout (Fig. S3), and that the lower boundary of the layer is warmer than the upper boundary throughout (based on the vertical boundary conditions in 2.1, also see example of temperature profile inset in Fig. 10), we can interpret the vertical terms as:

   a. heat is uniformly advected upwards through Ekman upwelling into the layer from below (SL: 0.7±0.3 Wm$^{-2}$, or a total of +0.9 ± 0.02 TW, and IC: 0.8±0.1 Wm$^{-2}$, or a total of 2.8 ± 003. TW).

   b. heat is diffused upwards out of the top of the layer, due to the relatively strong vertical temperature gradient at the thermocline, and due to vertical instabilities at the thermocline (SL: -2.3 ± 0.5 Wm$^{-2}$, or a total of -2.8 ± 0.2 TW, and IC: -2.1 ± 0.3 Wm$^{-2}$, or a total of -8 ± 0.4 TW). This heat may eventually be lost to the atmosphere, especially in the SL.

7. East of the Prime Meridian, 2 ellipses of strong heat flux divergence and 2 ellipses of strong heat flux convergence are found to the north and south of the eastern sub-gyre respectively. This is directly related to horizontal mean advection and is likely an artifact of a misalignment between horizontal circulation and strong horizontal temperature gradients on a coarse resolution grid. However, the "misalignment" may be linked to the occurrence of unresolved mesoscale eddies that are not represented by turbulent heat flux diffusion, that may be skewing the mean state representation of the temperature and flow fields. This is in part due to poor data coverage in a region where instabilities are likely generated from the interaction between "cold" regime recirculating WDW and incoming "warm" regime WDW (Gordon and Huber, 1984 and Fig. 1). Unresolved horizontal circulation around Maud Rise adds to the uncertainty in the region. Thus, to improve estimates in the eastern Weddell Gyre, more observations are required to resolve the complex ocean dynamics on smaller length-scales in the eastern sub-gyre region.

From using Argo floats, we have described the heat budget of a 1000 m thick layer encompassing the core of WDW within the Weddell Gyre. The role of mean horizontal advection is evident in feeding heat towards the southwestern Weddell Gyre, where the Filchner-Ronne ice shelves and Antarctic Peninsula are located. What is also important, however, is understanding the respective roles of mean horizontal advection and horizontal turbulent diffusion in removing some of that heat from the southern limb of the Weddell Gyre before it is able to reach the southwestern interior. This is crucial since Hellmer et al. (2012) suggest that under future climate scenarios, a redirection of the coastal current toward the Filchner-Ronne ice shelf could lead to increased advection of waters into the ice-shelf cavity, leading to increased basal ice melt from 0.2 to 4 m/year. While Naughten et al. (2021) argues that global temperatures would need to reach 7°C for warm water to intrude to the ice-shelf cavity, which exceeds the pledges in the Paris Agreement, nonetheless, the authors argue that unless global temperatures level out, melting of the ice shelf will at some point prevail.

**Data availability:** The Argo float data were collected and made freely available by the International Argo Program and the national programs that contribute to it (http://www.argo.ucsd.edu, http://argo.jcommops.org). The variable horizontal diffusivities were provided by Florian Sevellec (Sevellec et al., 2020: https://doi.org/10.17882/91335). The ERA5 dataset is available at https://www.ecmwf.int/en/forecasts/datasets/reanalysis-datasets/era5. Sea-ice data are available at https://nsidc.org/data/nsidc-0116/versions/4. Objectively mapped temperature for the time period 2002-2013 following the method used in this manuscript is available for download: Reeve et al. (2016): https://doi.pangaea.de/10.1594/PANGAEA.842876, and the 50-2000 m vertically integrated stream function for 2002-2016 derived from Argo floats is available for download at PANGAEA (Reeve et al., 2023; doi under review).

**Author Contribution:** K.A. Reeve performed the data analysis, figure preparation and wrote the manuscript with contributions from all authors.

**Competing interests**: The authors declare that they have no conflict of interest.

**Acknowledgements**: These data were collected and made freely available by the International Argo Program and the national programs that contribute to it (http://www.argo.ucsd.edu, http://argo.jcommops.org). The Argo Program is part of the Global Ocean Observing System. The GEBCO Digital Atlas is published by the British Oceanographic Data Centre on behalf of IOC and IHO, 2003. The mean velocities derived from mooring data were provided by Nicolas Le Paih, to whom the authors are indebted to. The variable horizontal diffusivities were provided by Florian Sevellec (Sevellec et al., 2020). All figures were created using MATLAB, in particular using the M_Map toolbox (Pawlowicz, 2020). KR is supported through the grant 424330345 of the Deutsche Forschungsgemeinschaft within the framework of SPP 1158 Antarktisforschung. The study also makes a contribution to EU SO-CHIC programme (grant number 821001) through the involvement of TK. This study is a contribution to the project T3 of the Collaborative Research Centre TRR 181 "Energy Transfers in Atmosphere and Ocean" funded by the Deutsche Forschungsgemeinschaft (DFG, German Research Foundation; project no. 274762653). MV was funded by the BMBF project APEAR (#03V01461). The authors are indebted to the anonymous reviewers whose feedback led to substantial improvement of the resulting analyses, figures, and manuscript, and also to Dr. Florian Sevellec, who also provided invaluable feedback regarding the approach and critical limitations to the heat budget computation.

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
