# Peer review of "The Weddell Gyre heat budget associated with the Warm Deep Water circulation derived from Argo Floats"

_EGUsphere, 2023_

## Referee Comment (RC2)

**The Weddell Gyre heat budget associated with the Warm Deep Water circulation derived from Argo Floats**

**Overall Assessment**

This study presents a heat budget of Warm Deep Water across the Weddell Gyre based on in situ Argo float data. Though the main results are not qualitatively surprising, they provide a valuable observations-based benchmark for the processes that distribute heat across the gyre. The main weakness of this analysis is that it relies on relatively sparse in situ data and crude parameterizations for unresolved eddy mixing. Though the authors are thorough in acknowledging the limitations of their analysis, certain key results remain insufficiently constrained. Nevertheless, with some revisions, this work will be a valuable addition to the literature.

My main criticisms are as follows:

- **Treatment of transient processes**: This study adapts the heat budget used by Tamsitt et al. (2016), who assessed zonal variations in heat fluxes along the Antarctic Circumpolar Current. Tamsitt et al. (2016) used five-day averaged output from the 1/6th-degree Southern Ocean State Estimate (SOSE). In this framework, "turbulent diffusion" has a clear physical interpretation. Since SOSE resolves time-mean and transient variations in the large-scale flow and mesoscale eddies, "turbulent diffusion" describes unresolved, subgrid-scale mixing. Here, the underlying dataset only provides (smoothed) time-averaged temperature and horizontal velocities, and all other processes (not counting vertical advection) are implicitly parameterized as "turbulent diffusion". While it may be reasonable to characterize eddy stirring as a diffusive process, it is problematic to treat transient variations in the large-scale circulation and temperature field in the same manner. There needs to be a more careful treatment of the heat fluxes associated with temporal correlations in the temperature and velocity field. The discussion in Section 3.1 should distinguish time-averaged from transient processes (i.e., via Reynold's decomposition) and clarify that only the former is resolved. If transient processes cannot be considered negligible, they should be treated as residuals rather than lumped with turbulent diffusion.

- **Representation of eddy-mixing and turbulent diffusion**. To account for unresolved eddies, the authors assume their net effect on the time-mean heat budget can be parameterized as a diffusive process. While this is a reasonable and standard approach, I am unconvinced that the effects of eddy mixing are within the bounds of uncertainty presented here. I particularly question the validity of the heat budget analysis east of Maud Rise, where numerous studies have demonstrated that mesoscale eddies have leading order control of heat transport in this region (Ryan et al. 2016, Wilson et al. 2022). Since these eddies mix water-mass properties on sub-seasonal scales and create spatial gradients much smaller than the smoothing filter used to create the temperature climatology, it is no surprise the heat budget does not come to closing in this area (Figs. 2-5). While it is useful to see Figure 2 in its current form, the subsequent analysis should be limited the analysis to the open-ocean areas east of Maud Rise, where the horizontal temperature gradients are weak and the eddies are not as energetic. In my opinion, the data are insufficient to provide a valid heat budget elsewhere. Additionally, I would like to see stronger

acknowledgment in the Abstract that the effects of eddy mixing are highly uncertain.

- **Over-reliance on arbitrary and ad hoc methods**: While budget analyses like this study inevitably involve some amount of arbitrary decisions, this study does so to an excessive extent. For example, in line 188, the authors arbitrarily define an uncertainty range for the diffusivity coefficients, and no rationale is given beyond the unsupported claim that these values are sufficiently large. Another example is when the authors split the Weddell Gyre into "interior cell" and "southern limb," where the former is eventually subdivided into northern and southern limbs. There is no clear rationale for why this is done, and the differences among these regions are sensitive to how the parts are defined. For the last example, if the goal is to highlight these meridional variations in the heat budget balance, a cleaner approach would be to compute a zonal average of the budget terms west of Maud Rise. I document other instances of these arbitrary and sometimes perplexing methodologies below.

**Detailed comments:**

- Lines 25-: The plain language summary reads more like a second abstract. I think this needs to be more concise and less technical.

- Line 34: "Warm Deep Water, however, varies in its properties too strongly to tease..." It is unclear what "varies too strongly" means.

- Line 39: "interesting features..." Please replace "interesting" with a more objective adjective.

- Line 53: "The CDW that enters the Weddell Gyre is commonly referred to as Warm Deep Water (WDW)..." This is a nitpick, but I understand WDW to be a modified variant of CDW rather than simply CDW that exists in the Weddell Sea.

- Figure 1: Add contour labels for the streamlines, specifically the ones used to define IC SL subregions.

- Line 152: Please briefly state how Sevellec et al. (2022) obtained their diffusivity estimates.

- Lines 135-138: A couple of things here:
    - Figure S1a and a summary of the accompanying discussion regarding the definition of the vertical boundaries of WDW should be included in the main text.
    - For Figure S1a, it would be helpful to include additional profiles to illustrate the variability of temperature profiles and the location of the upper boundary.
    - Regarding the previous point, are there regions where the lower boundary temperature is cooler than the upper boundary temperature?

- Line 154: add "a" between acknowledging and lack.

- Lines 203-205: Please provide a more physical motivation for defining the IC and SL regions. These seemingly arbitrary definitions undermine the robustness of these results.

- Lines 254: I would rephrase "useful information" more objectively and state specifically why we should trust the spatially averaged values when the local details are not considered reliable.

- Figure 3: Apologies if I missed this in the text, but what fraction of $A_H$ goes north versus southward to the shelf?

- Line 319: I would argue that the budget does not close anywhere in the domain.

- Lines 323-324: It is odd to disregard the easternmost values in this section and not elsewhere. More consistency is needed. See my third major comment.

- Line 494: I am not sure what "ellipses" refer to.

- Lines 510-513: I suspect unresolved mesoscale eddies have a leading order impact on the mesoscale heat budget. In addition to the observational studies referenced in the following sentence, idealized modeling studies indicate that transient eddies (e.g., Wilson et al. 2022) are responsible for most of the southward heat transport in the eastern limb of the gyre.

- Line 543: To be more precise, there is a Taylor Cap rather than a Taylor Column over Maud Rise. It is also inaccurate to say that the water column above the Rise is "stagnant" since it does exchange water mass properties with the ambient fluid.

- Figure 8: This is a lovely summary figure.

**References:**

Ryan, S., Schröder, M., Huhn, O., and Timmermann, R.: On the warm inflow at the eastern boundary of the Weddell Gyre, Deep-Sea Res. I, 107, 70-81, https://doi.org/10.1016/j.dsr.2015.11.002, 2016.

Wilson, E. A., A. F. Thompson, A. Stewart, and S. Sun (2022), Bathymetric control of the subpolar gyres and overturning circulation in the Southern Ocean, Journal of Physical Oceanography, 52, 205–223, https://doi.org/10.1175/JPO-D-21-0136.1.

---

## Author Comment (AC1)

**Reviewer 1**

In this work, the authors carry out a heat budget analysis in the Weddell Gyre using a suite of observations, including a gridded climatological product that covers 2002-2016. They find that the budget is very noisy on the grid scale, with local errors that are sometimes as large as the quantity being estimated. However, the authors claim that integrating over sufficiently large spatial scales allows them to make quantitative statements about the relative importance of advection and diffusion in both the horizontal and vertical directions. Specifically, they find that in the interior of the Weddell Gyre, net warming from horizontal turbulent heat diffusion is largely balanced by net cooling from mean horizontal advection. In contrast, in the southern limb of the Weddell Gyre, the relative contribution of these two terms switch signs, i.e., net cooling from horizontal turbulent heat diffusion is largely balanced by net warming from mean horizontal advection. As far as I am aware, this key result is novel, and it is certainly relevant to our understanding of the Weddell Gyre heat budget.

Observational analysis in the Weddell Gyre is extremely challenging due to the sparsity of measurements from that remote, harsh part of the world. The authors have managed to extract some novel estimates from these sparse observations and noisy heat budget by integration. The paper is very thorough and detailed, and it contributes to a growing body of work that the lead author has been assembling in recent years.

I do have some concerns that mostly have to do with how some of the results are presented. Although the authors have been up front about the errors in their estimates, I believe the errors should be emphasized even more in certain plots (e.g., the spatial maps). For example, when comparing Fig. 2a with Fig. S8, we see total propagated errors that are at least as large as the estimates themselves, although the saturated color bars make it difficult to compare in some regions. It is not clear to me whether some of the features in the estimate maps that are discussed in the text (e.g., the bands of alternating convergence and divergence) are robust, given the size of the propagated errors. Overall, after some edits for clarification, this manuscript will make a valuable addition to the literature; I recommend that this article be returned for minor revisions. I have included some specific suggestions below; I hope that they are helpful.

> We would like to thank the reviewer for the constructive and very positive feedback, and for taking the time and effort to go through this rather long paper. We have followed your suggestions and we feel they have resulted in a much-improved manuscript. In addition, we have made some key changes to the manuscript, in consideration of the comments here and also of the other reviewer. We summarise these here briefly before responding to your individual comments. We provide the tracked changes manuscript to highlight these changes, as well as pdf file of all figures, captions and tables, for easy viewing for the reviewer. Line numbers refer to the line numbers of the changes in the manuscript with tracked changes.

> The three major changes that we have made are as follows. Instead of arbitrarily defining the diffusivities within the Weddell Gyre (i.e., where data from Sevellec et al. are unavailable), we directly use diffusivities provided by Donnelly et al. (2017), given these are derived from observations throughout the Weddell Gyre. Thus, $\kappa_H$ is now 247±63 m$^2$s$^{-1}$ (the error range also coming from the paper) and $\kappa_V$ is now $(2.39 \pm 2.83) \times 10^{-5}$ m$^2$ s$^{-1}$ (lines 185-188). We think this simplifies things and is also more representative of the gyre. We also include the approximate locations of the ship stations from Fig 3 in Donnelly et al (2017) in Fig. S4, showing the maps of $\kappa_H$.

> Secondly, we changed the zonal analysis to focus on the regions west of Maud Rise, at 3°E (line 308). This is due to the high uncertainties in the eastern Weddell Sea, largely due to unresolved mesoscale

eddy processes. This means that all the figures are new in section 4.2. The mean and net heat fluxes for the different terms are thus different, but the overall interpretation of the results remain unchanged. The text has been altered, where descriptions east of Maud Rise were removed. We also provide a zonal and meridional mean and integration of the whole western Weddell Gyre region (i.e., IC + SL) in the supplements. These show the same results as separate regional analyses in section 4.2, which is why they are kept in the supplements (section S8).

Lastly, we altered our definition of the SL region (line 298). Initially, we focused on the region where the streamlines extend zonally across the whole extent of the gyre. However, by defining it in this manner, there was a slight overlap with the IC region, which resulted in ambiguous interpretations of the results. Thus, we now define SL as the open inflow region, where the streamlines still span the full zonal extent of the Weddell Gyre, where the water masses entering the open inflow zone are entering from outside the gyre from the east. The IC, in contrast, is still defined by a streamline forming a fully enclosed circuit, and thus focuses on recirculation water masses within a true gyre structure (Fig. 4). Again, this results in a slight change to the mean and net heat fluxes (interestingly resulting in a closure of the SL heat budget, but not of the IC), but the overall interpretation remains the same.

**Text**

Line 55: Change to "en route" (it's a phrase borrowed from French)

Done (line 69)

Line 71: Is the seasonal cycle "unresolved", or has it been excluded in your analysis in this paper? Here it's not clear whether you are saying that Argo does not resolve the seasonal cycle, or if you have excluded it by discarding everything above your dynamic upper boundary. Please clarify in the text.

The seasonal cycle has indeed been excluded, due to inadequate data density. When considering smaller regions where data is present (e.g., individual floats or small groups of floats), one could observe the seasonal cycle. We are, however, attempting to understand the broader gyre-scale perspective and therefore lack the data to incorporate a seasonal analysis into our optimal interpolation. Therefore, we exclude the upper 50 m. Adjusted the sentence as follows:

Line 85: *"Given Argo float data lacks the spatial-temporal coverage to resolve the seasonal cycle while also objectively mapping the entire region, a full basin analysis of the upper 50 m is unfeasible."*

Line 130: Shouldn't this be "the rate of change of heat storage of a certain ocean volume"? It's the rate of change that is affected by the net flux.

Good point, changed accordingly, thanks (line 145)

Lines 73-74: Is this elongation of the "shape of the area of influence" a consequence of how the objective mapping was calculated? Please clarify in the text: why does the f/H fractional scale alter the shape of the area of influence?

The "shape of the area of influence" is a result of the e-folding decay scale of the covariance function (details in Reeve et al., 2016), which is determined by both the horizontal length scale and the difference in f/H between the grid point one is interpolating to, and the data points themselves. E.g., say you have 2 datapoints, equidistant to the grid point, but one has a similar f/H to the grid point, whereas the other

has a much shallower (or deeper) water column depth, H (assuming close proximity of data point to grid point limits the impact of changing f, which suggests that bottom bathymetry variation plays a larger role). Then the data point with a more similar f/H will carry a larger weight. Whereas horizontal length-scales alter the size of the "area of influence", scaling the covariance with f/H impacts its shape. A nice demonstration of this can be found in Fig. 9 in Reeve et al., 2016. This is to reflect the tendency of water masses to conserve potential vorticity.

Altered sentence as follows:

Line 209: *"The fractional scale on the effect of f/H allows us to take into account the tendency of water parcels to follow paths of constant f/H in order to conserve vorticity (for a visual demonstration of the impact of including f/H in the scaling, see Fig. 9 in Reeve et al. 2016)."*

Lines 202-206: I suggest moving this detailed description to the start of section 4.2, just before you start discussing it.

Done. Last part of section now reads:

Line 240: "*whereas in part 2, we consider the larger-scale zonal variation of the heat budget*"

And the beginning of Section 4.2 now reads:

Line 298: "*In this section, we consider the zonal variation of the heat budget, for two regions: (1) the open southern limb (SL) and (2) the interior circulation cell (IC). The open SL region (i.e., the magenta stippled area in Fig. 4) is defined by the stream function as $16 \leq \Psi \leq 26$ Sv, which describes the open inflow zone where water masses enter the gyre, and spans the entire zonal extent of the double gyre system, from just west of Gunnerus Ridge (~33°E) to ~50° W, where the streamlines veer northwards to follow the coastline of the Antarctic Peninsula. The southern boundary of the SL is the southernmost streamline that does not intersect with the coastline (16 Sv). This definition of the SL enables us to focus on the water that enters the gyre from the east, and circulates the entire zonal extent of the gyre, thus reaching into the south-western interior. The IC region (i.e., the blue stippled area in Fig. 4) is defined as $\Psi \geq 26$ Sv, which is the largest streamline that spans the entire zonal extent of the double gyre system, this time forming a fully enclosed circuit. This definition of the IC allows us to focus on the recirculating waters of the gyre, from just west of Gunnerus Ridge to near the continental shelf edge of the northern tip of the Antarctic Peninsula (~50° W). For both regions the area east of Maud Rise (3° E) is omitted, due to large uncertainties east of Maud Rise (discussed in Section 5)."*

Suggestion for new figure: I would suggest adding a new, separate figure to show your two analysis regions of integration in a more visually intuitive. At present, the depictions of the regions are incorporated into an already-busy figure, and the blue contour risks causing some confusion about the extent of the regions, even with the stippling (e.g. the portion of a blue contour in the bottom right of Fig. 2d is distracting). The new figure could have isobaths, streamlines, and stippling to show exactly where the two regions exist, with no extra information competing for attention.

We have now added the figure as you suggested (Fig. 4), thank you for the suggestion. We also altered our definition of the SL region slightly to focus only on the open inflow (before there was a slight overlap with the IC region), where the streamlines imply an inflow of water from the east, outside the gyre. We've done this in order to avoid ambiguity in our interpretation of the relationship between the SL and IC regions. This has resulted in a change in the numbers, but not the overall findings (and the heat budget even comes to a close in the SL region now (0.3 ± 3 TW, or 0.002 ± 0.02 °C/yr).

Lines 211-212: Why is this "small patch of divergence" being singled out? Its significance is not obvious, and given the large size of the propagated errors (Fig. S8), it's not clear if these features are robust.

> We agree with the reviewer on this and have removed the statement and the subsequent sentence, and replaced them with the following:

> Line 249: "*While small areas of heat flux divergence are found throughout the southern limb, these small areas appear to be less prominent than the areas of positive heat flux and are unlikely to be significant given the high associated errors (Fig. 3a, right panel)."*

Section 4.1 overall: As discussed above, the large propagated errors calls the robustness of these features into question. It's not clear if detailed interpretation of these features, especially ones with small magnitudes, tells us very much. By comparison, the results that come out of the later large-scale integrations seem more robust. I suggest de-emphasizing some of the discussions of specific, detailed features.

> Thank you for the suggestion. We deleted the feature-specific descriptions as shown in the previous response. We limited the description to generalized broad areas such as east vs west. We also deleted a sentence describing slightly enhanced vertical turbulent diffusion on line 277. We also include the error maps alongside the maps of the terms, in Fig. 3. As per your suggestion under figures and captions.

Line 224-225: It does imply that the net effect of the vertical advection is warming. The second part of the phrase could be somewhat confusing. I suggest rephrasing this as "…through the top, implying that the net effect of vertical advection is to warm the layer."

> Done (line 266)

Line 242, 245: Replace "chapter" with "section".

> Done

Line 342-344: Change to a separate sentence, i.e. "There is an overall decrease in temperature along the southern limb of the Weddell Gyre, and an overall increase in temperature along the northern limb (Fig. 6)." That way the order of the text matches the order of someone reading the figure left-to-right.

> Done (line 416)

Line 458: Is the vertical grid uniform? A changing vertical grid could also introduce a spatial scale bias in the estimates.

> This is a good point. The vertical grid is not uniform, with highest resolution in the shallower part of the water column. We take the upper boundary as the mean across neighboring grid points, and assume that the associated bias would be minor in comparison to the vertical boundary conditions themselves. We added the following in section 5.1.1 on vertical boundary limits:

> Line 453: "However, this, along with the non-uniform resolution of the vertical grid, may introduce some noise into the analysis from grid cell to grid cell, owing to the different depths of the water column the heat budget is integrated over."

Line 478: Perhaps change "negative air-to-sea heat flux" to "radiative heat loss" for simplicity?

*Done (line 557)*

Section 5.2.2: This section is very thorough, but it is very long and cumbersome. If allowed by EGU formatting, it could benefit from paragraph headers (sometimes called inline headings) to help orient the reader. Alternatively, it could be section 5.3, broken up into smaller sub-subsections.

*We have now added inline headings to this section, thank you for the suggestion.*

Lines 483-484: This statement doesn't work as a standalone phrase. Perhaps rephrase as "

*The reviewer's suggestion for a rephrase was missing above. While we are still open to suggestion, in the meantime we have rephrased the sentence as follows:*

*Line 562: "The heat budget shown in section 4 does not close, especially when integrated over smaller areas"*

**Figures and captions**

Figure 1: The red stars are very difficult to see. Perhaps make the markers much bigger and change the color so they stand out better? Figure 1 caption: Isn't "conservative temperature at the depth of the temperature maximum" just "maximum temperature?"

*Done, and altered the caption to "sub-surface conservative temperature maximum"*

Figure 1 caption: Change to "(1) Gunnerus Ridge, (2) Astrid Ridge, and (3) Maud Rise" instead of using the "respectively" construct.

*Done*

Figure 2: This figure risks being unintentionally misleading. If I have understood correctly, on a local scale, the propagated error is often just as large as the estimate itself, which throws the robustness of these spatial patterns into question, patchy as they may be. There is a risk of over-interpreting features, as they may reflect biases in the estimates. The authors have been very up-front and thorough about these biases in the manuscript overall, but casual readers may get the wrong idea glancing at these figures. I'd suggest moving the spatial patterns to the supplemental information and focusing on the integrals, and/or putting the propagated errors in their own column next to the mean terms for easy comparison.

*The authors agree with the reviewer here, and decided to move the propagated errors to a column alongside the main terms, in Fig. 3. In reference to your earlier comment (i.e., 3[rd] paragraph, stating concerns about associated errors to the alternating bands of heat flux convergence and divergence in Fig. 3a), the large errors sit directly north to these alternating bands. We added the following to the text on line 256:*

*"Note associated errors become increasingly large directly to the north of the northern limb, related to the highly dynamic boundary."*

Figure 3 caption: Explain the meaning of the shaded areas here in the caption, since this is where they are first introduced in the figures.

*Added the following to Fig. 5 caption: "The shaded errors provide the associated propagated error (detailed in section 3.2 and the supplement)."*

Figure 6: I would suggest making this figure even more visually obvious – consider that people may only glance at it briefly. You could add a dashed vertical line separating the northern limb from the southern limb. On their side of that line, you could add a text box with either "northern limb" or "southern limb", just to make it explicit and clear. I'd also suggest using a different symbol for 30°W (north) and 30°W (south).

Thank you for the useful suggestions. We hope the reviewer finds now Fig. 8 much improved.

Order of Fig. 5, 6, and 7: Perhaps figure 6 should come before figure 5? You could introduce the temperature tendency first and then show the integrated budget terms afterwards, as a way of explaining these temperature trends.

We understand the logic of the suggestion and gave this a go but it didn't really work with the restructuring of the text, since this is specifically relating to the inverse relation between horizontal advection and turbulent diffusion in Fig. 7a. Thus, we have decided to leave it as is.

Figures 5 and 7: Please add y-axis labels for every panel, as the unit by itself is not sufficient. What quantity are we looking at?

Done

Figure 8: The inset figure appears to be distorted, and the y-axis labels have been cut off. As for the main figure, I suggest using a legend to help quickly orient the reader as to the meaning of the symbols and numbers. At present, trying to understand the figure from the caption is visually overwhelming. You may also want a simple figure with a legend for talks. If you get this figure right, lots of people will show it in their own talks, summarizing your important results.)

Thank you for the positive feedback and useful suggestion. We have now provided a legend which summarises the content of the original caption, and subsequently shortened the caption by half (now Fig. 10).

**Additional references**

I would suggest adding two references to the paper, if the authors agree that they are suitable. (Note: I was not involved with either of these papers.) First, there is a recent numerical modeling study that explored the seasonal and interannual variability of the Weddell Gyre in a high-resolution model. I suggest that the authors mention this in the discussion section; it is a concrete illustration of seasonal and interannual biases, which are relevant to this climatology.

Neme, J., England, M. H., & Hogg, A. M. (2021). Seasonal and interannual variability of the Weddell Gyre from a high-resolution global ocean-sea ice simulation during 1958–2018. Journal of Geophysical Research: Oceans, 126, e2021JC017662. https://doi.org/10.1029/2021JC017662

Second, the authors end the paper by mentioning the vulnerability of the Filchner-Ronne ice shelf. There is a more recent reference that explores the tipping point beyond which the cavity beneath the Filchner-Ronne Ice Shelf will flood with warm water:

Naughten, K.A., De Rydt, J., Rosier, S.H.R. et al. Two-timescale response of a large Antarctic ice shelf to climate change. Nat. Commun. 12, 1991 (2021). https://doi.org/10.1038/s41467-021-22259-0

Thank you for the recommendations and for pointing out these interesting papers. We have added the following:

Line 590 (section 5.2.2): "*The summertime bias and coarse resolution likely results in an underestimate of the mean gyre strength, and thus the velocity field derived from the stream function (Neme et al., 2021).*"

And:

Line 805: "*While Naughten et al. (2021) argues that global temperatures would need to reach 7°C for warm water to intrude to the ice-shelf cavity, which exceeds the pledges in the Paris Agreement, nonetheless, the authors argue that unless global temperatures level out, melting of the ice shelf will at some point prevail.*"

---

## Author Comment (AC2)

**Reviewer 2:**

**Overall Assessment**

This study presents a heat budget of Warm Deep Water across the Weddell Gyre based on in situ Argo float data. Though the main results are not qualitatively surprising, they provide a valuable observations-based benchmark for the processes that distribute heat across the gyre. The main weakness of this analysis is that it relies on relatively sparse in situ data and crude parameterizations for unresolved eddy mixing. Though the authors are thorough in acknowledging the limitations of their analysis, certain key results remain insufficiently constrained. Nevertheless, with some revisions, this work will be a valuable addition to the literature.

> We would like to thank the authors for the constructive criticism of the manuscript. We found the points raised and the suggestions to be extremely valuable, helping us to improve the overall manuscript. We would also like to thank them for taking the time to go through the paper in depth – we are very aware it's not the shortest paper. We hope we have answered and satisfied the concerns of the reviewer and will detail the key changes we have made throughout our responses to each point below. We provide the tracked changes manuscript to highlight these changes, as well as a pdf file of all figures, captions and tables, for easy viewing for the reviewer. Line numbers refer to the line numbers of the changes in the manuscript with tracked changes.

**My main criticisms are as follows:**

- **Treatment of transient processes**: This study adapts the heat budget used by Tamsitt et al. (2016), who assessed zonal variations in heat fluxes along the Antarctic Circumpolar Current. Tamsitt et al. (2016) used five-day averaged output from the 1/6th-degree Southern Ocean State Estimate (SOSE). In this framework, "turbulent diffusion" has a clear physical interpretation. Since SOSE resolves time-mean and transient variations in the large-scale flow and mesoscale eddies, "turbulent diffusion" describes unresolved, subgrid-scale mixing. Here, the underlying dataset only provides (smoothed) time-averaged temperature and horizontal velocities, and all other processes (not counting vertical advection) are implicitly parameterized as "turbulent diffusion". While it may be reasonable to characterize eddy stirring as a diffusive process, it is problematic to treat transient variations in the large-scale circulation and temperature field in the same manner. There needs to be a more careful treatment of the heat fluxes associated with temporal correlations in the temperature and velocity field. The discussion in Section 3.1 should distinguish time-averaged from transient processes (i.e., via Reynold's decomposition) and clarify that only the former is resolved. If transient processes cannot be considered negligible, they should be treated as residuals rather than lumped with turbulent diffusion.

> Thank you for making this very good point. We have done our best to address these issues by acknowledging the limitations of the study. We now include the following in section 3.1:
>
> Equations 1.1 and 1.2 now include a 5th term, *R*, and added the following in line 150:
>
> "*R represents the unresolved transient processes excluded in this study (discussed below).*",
>
> followed by the following more in-depth discussion:
>
> *Line 168: "Note that given the constraints of the method used (i.e., our data resources are an objectively mapped long-term mean temperature field, and horizontal velocity derived from a long-term mean stream function of the Weddell Gyre, derived from in situ observations), we are unable to look at deviations from the mean, i.e., transient processes. This means, that the meaning of advective and diffusive heat fluxes are different from the ones quantified by Tamsitt et al. (2016). Their underlying numerical model resolves large-scale variations of the flow and temperature field and (partly) mesoscale eddies; these processes are part of the advection, while turbulent diffusion refers to unresolved small-scale processes. In our study, advection is computed from time mean quantities while the effects of mesoscale eddies are parameterised by horizontal turbulent diffusion. Large-scale variations of the flow field are not accounted for in our study. Thus, we have an additional, unknown 5th term in the heat budget in Eq. 1.1*

*and 1.2, R, which represents the unresolved transient processes excluded from the study. Increased spatial and temporal coverage of in situ observations within the Weddell Gyre would be required to address these gaps (further discussed in section 5.2.2)."*

We also discuss these limitations and attempt to make them clearer in the discussion:

Line 597: *"This is a key consideration, which means that the sum of the 4 heat budget terms, especially in the east, cannot be viewed as the heat tendency (which should be zero in a closed system), but that it additionally consists of unresolved processes (i.e., "R" in Eq. 1.1 & 1.2). These include mesoscale eddy activity that is not fully represented by parameterization via turbulent diffusion. We know that the eastern part of the eastern sub-gyre is dominated by an intense mesoscale eddy field (Ryan et al., 2016, Leach et al., 2011 and Gordon and Huber 1984). Wilson et al. (2022) show, using idealised models, that transient eddies are responsible for most of the southward heat transport in the eastern limb of the Weddell Gyre. In addition, as discussed in section 3, a process not accounted for is regional-scale variations in the temperature and flow fields. This process might be particularly important in the eastern sub-gyre, where the boundary to the gyre is poorly-defined due to the openness of the topography. Indeed, Schröder and Fahrbach (1999) suggest that there is no continuous current marking the eastern boundary, and that baroclinic shear instabilities lead to a breakdown of the eastward-flowing current in the northern limb of the gyre, and that the current "reforms" in the westward-flowing southern limb. The geometry of the eastern sub-gyre might therefore be sensitive to interannual to decadal variations in the wind forcing, potentially affecting the time mean heat flux convergence in this area."*

Line 611: *"In the eastern sub-gyre region, recirculated "cold-regime" WDW (modified primarily through heat loss) comes into contact with incoming "warm-regime" WDW (Gordon and Huber, 1984). The "warm-regime" WDW represents relatively warm WDW advected into the gyre at the eastern inflow zone at about 30° E, driven by mesoscale eddies (Deacon 1979; Orsi et al. 1993; Orsi et al. 1995; Gouretski and Danilov 1993, 1994; Ryan et al., 2016). When comparing the two terms in Fig. 3a and 3c, while the magnitude is much larger in Fig. 3a (horizontal mean advection), horizontal turbulent diffusion displays the opposite signs and partially compensates in the eastern sub-gyre region in Fig. 3c. The terms do not cancel, and thus imply that the missing R term is significant in this region (although, large errors associated with mean horizontal advection also imply that mean advection is poorly represented in this region). This is not easily remedied, since the eastern Weddell Gyre is a region with poor data coverage, including from Argo floats, though at the time of writing, efforts are underway to close this key observational gap."*

Line 625: *"We'd like to acknowledge that our framework of inferring the heat budget is rather traditional, in which we parameterize the effects of eddies by means of horizontal diffusion. A more advanced approach is represented by the temporal-residual-mean framework (McDougall and McIntosh, 2001), in which the effects of eddies are decomposed into eddy-induced advection (adding an eddy-induced velocity to the time mean velocity in the advection term of the tracer equation) and eddy-induced diffusion. The latter can be decomposed into isopycnal and diapycnal diffusion (Groeskamp et al., 2016). We acknowledge this framework to be more physics-based than our classical approach, yet, given the limitation of our dataset, the estimation of the eddy-induced velocities is not straightforward. At the same time, Sevellec et al. (2019) demonstrated the usefulness of the temporal-residual-mean framework when applied to interpreting eddy-driven horizontal buoyancy transports from mooring-based observations acquired in Drake Passage. In particular, they highlight importance of eddy-driven horizontal transports in the direction perpendicular to the mean flow. For future work it would therefore be intriguing to demonstrate, whether the application of this framework to our data set would represent a major step forward towards closing the heat budget in the eddy-rich eastern part of the Weddell Gyre and around Maud Rise, where our approach does not lead to satisfactory results."*

- ***Representation of eddy-mixing and turbulent diffusion***. To account for unresolved eddies, the authors assume their net effect on the time-mean heat budget can be parameterized as a diffusive process. While this is a reasonable and standard approach, I am unconvinced that the effects of eddy mixing are within the bounds of uncertainty presented here. I particularly question the validity of the heat budget analysis east of Maud Rise, where numerous studies have demonstrated that mesoscale eddies have leading order control of heat transport in this region (Ryan et al. 2016, Wilson et al. 2022). Since these eddies mix water-mass properties on subseasonal scales and create spatial gradients much smaller than the smoothing filter used to create the temperature

climatology, it is no surprise the heat budget does not come to closing in this area (Figs. 2-5). While it is useful to see Figure 2 in its current form, the subsequent analysis should be limited the analysis to the open-ocean areas east of Maud Rise, where the horizontal temperature gradients are weak and the eddies are not as energetic. In my opinion, the data are insufficient to provide a valid heat budget elsewhere. Additionally, I would like to see stronger acknowledgment in the Abstract that the effects of eddy mixing are highly uncertain.

> We agree with the authors and have now computed the zonal means and integrations to focus solely on west of Maud Rise, as suggested by the reviewer. The upper panels in the figures 5-7 & 9 show the zonal mean heat budget terms in Wm$^{-2}$. Overall, the numbers are changed (the net terms are obviously smaller due to a smaller area coverage), but the interpretation of the results remains unchanged. Indeed, the sum of the heat budget terms closes within uncertainty bounds for the SL (0.3 ± 3 TW). The heat budget still does not close for the IC, due to the influence of high diffusivity values from the Sevellec et al. dataset (Fig. S4, 3c, 6). We have also edited the text, to remove any descriptions of analyses east of Maud Rise which are no longer included, and added text to describe the new analysis and emphasise the importance of eddying processes. For example,
>
> *Line 470: "Unresolved eddying fluctuations are most likely to be an important factor when assessing the heat budget east of the Prime Meridian in the Weddell Gyre, a region we know to be dominated by a mesoscale eddy field (e.g., Schröder & Fahrbach, 1999; Leach et al., 2011; Ryan et al., 2016). We therefore exclude the region east of Maud Rise in the regional analysis in section 4.2."*
>
> We also added the following sentences to the Abstract:
>
> Line 14: *"While the results are somewhat noisy on the grid scale, and the representation of the effects of eddy mixing is highly uncertain due to need to parameterize them by means of turbulent diffusion, the heat budget (i.e., the sum of all terms) closes (within the uncertainty range) when integrated over the open inflow region in the southern limb, whereas the interior circulation cell remains unbalanced"*
>
> Line 28: *"Temporal deviations from the mean terms are not included due to study limitations. In order to appreciate the role of transient eddying processes, a continued effort to increase the spatial and temporal coverage of observations in the eastern Weddell Sea is required."*

- ***Over-reliance on arbitrary and ad hoc methods***: While budget analyses like this study inevitably involve some amount of arbitrary decisions, this study does so to an excessive extent. For example, in line 188, the authors arbitrarily define an uncertainty range for the diffusivity coefficients, and no rationale is given beyond the unsupported claim that these values are sufficiently large. Another example is when the authors split the Weddell Gyre into "interior cell" and "southern limb," where the former is eventually subdivided into northern and southern limbs. There is no clear rationale for why this is done, and the differences among these regions are sensitive to how the parts are defined. For the last example, if the goal is to highlight these meridional variations in the heat budget balance, a cleaner approach would be to compute a zonal average of the budget terms west of Maud Rise. I document other instances of these arbitrary and sometimes perplexing methodologies below.

> 1. Diffusivity: We acknowledge the diffusivity is arbitrarily defined, based on values in the literature. This is a key component of the paper that the authors hope to rectify in future research by using tracked floats within the gyre to estimate diffusivity. We have now decided to define the $\kappa_H$ and $\kappa_V$ values based on findings from Donnelly et al (2017) given the results are thorough and provide estimates for within the Weddell Sea for both vertical and horizontal, and the authors do a thorough comparison with other estimates in the literature. Thus, $\kappa_H$ is now 247±63 m$^2$s$^{-1}$ (the error range also coming from the paper) and $\kappa_V$ is now (2.39 ± 2.83) × 10$^{-5}$ m$^2$ s$^{-1}$. We think this provides a justified foundation to the study and is also more representative of the gyre, especially west of Maud Rise. We also include the approximate locations of the ship stations from Fig 3 in Donnelly et al (2017) in Fig. S4a, showing the map of $\kappa_{Hx}$.

To explain our choice, we have added on Line 185:

*"For the remainder of the Weddell Gyre, we define $\kappa_H$ as $247 \pm 63$ $m^2s^{-1}$ and $\kappa_V$ as $(2.39 \pm 2.83)$ $x$ $10^{-5}$ $m^2s^{-1}$ based on the estimates provided by Donelly et al. (2017), which are derived from ship-based observations throughout the Weddell Sea in combination with velocity estimates from the ECCO live access server."*

2. Sub-regions: We did provide a rationale on how we defined the sub-regions. We define our regions based on our knowledge of the mean horizontal circulation. This is done by using the stream function, where we focus on the eastward flowing northern limb and the westward flowing southern limb. Originally, we defined the SL as the area as indicated by streamline where the westward flow extends the full zonal extent of the gyre, whereas the IC is defined as the region fully enclosed by a single streamline (i.e., a "true" gyre circulation). This, however, resulted in a slight overlap between regions, which we acknowledge may lead to ambiguity regarding our interpretation of the results. Thus, we have now defined SL as the open inflow region, where the streamlines indicate a westward inflow from outside the gyre in the east, all the way to the western interior, whereas the IC is still defined by its single, fully enclosed streamline, and is thus representative of recirculating water masses.

In order to make our choices more transparent, we have now added a map clearly marking these two regions in Fig. 4, as well as clearer explanations of our subregions:

Line 298 (opening to Section 4.2): *"In this section, we consider the zonal variation of the heat budget, for two regions: (1) the open southern limb (SL) and (2) the interior circulation cell (IC). The open SL region (i.e., the magenta stippled area in Fig. 4)) is defined by the stream function as $16 \le \Psi \le 26$ Sv, which describes the open inflow zone where water masses enter the gyre, and spans the entire zonal extent of the double gyre system, from just west of Gunnerus Ridge ($\sim33°E$) to $\sim50°$ W, where the streamlines veer northwards to follow the coastline of the Antarctic Peninsula. The southern boundary of the SL is the southernmost streamline that does not intersect with the coastline (16 Sv). This definition of the SL enables us to focus on the water that enters the gyre from the east, and circulates the entire zonal extent of the gyre, thus reaching into the south-western interior. The IC region (i.e., the blue stippled area in Fig. 4) is defined as $\Psi \ge 26$ Sv, which is the largest streamline that spans the entire zonal extent of the double gyre system, this time forming a fully enclosed circuit. This definition of the IC allows us to focus on the recirculating waters of the gyre, from just west of Gunnerus Ridge to near the continental shelf edge of the northern tip of the Antarctic Peninsula ($\sim50°$ W). For both regions the area east of Maud Rise ($3°$ E) is omitted, due to large uncertainties east of Maud Rise (discussed in Section 5)."*

3. The reviewer suggested that we compute the zonal average of the heat budget terms to highlight the meridional variations.

We have also now computed the zonal and meridional means and integrations for the whole Weddell Gyre region west of Maud Rise. These figures are shown below. We find these new results to be a useful validation of our regional analysis. We have therefore included them and the description above in the supplements (Section S8, Figs. S7-8). This is because the regional analysis focuses on a spatial analysis of the heat distribution and is more comprehensible in terms of the analysis which leads to our final schematic in Fig. 10. We add the following in the opening paragraph of section 4.2, line 324:

*"We also provide a zonal and meridional analysis of the entire region marked by both blue and magenta stippling in Fig. 4 in the supplements (Figs. S7 and S8). These analyses provide results that agree with the analyses presented in this section and are described in section S8."*

And the following description in the supplements, section S8:

*"The zonal and meridional means and integrations for the whole Weddell Gyre region west of Maud Rise (i.e., SL + IC) were also computed for the heat budget terms. These figures are shown below. The zonal mean in Fig. S7 is very similar to the IC analysis, although it shows a net zero contribution of mean advection, which makes sense because of the gyre-characteristics of the circulation, where westward flowing southern and eastward flowing*

*northern limbs cancel with each other: heat is advected into the gyre east of 10 °E, and then advected out of the gyre west of 40 °W. Horizontal turbulent diffusion dominates as a heat source east of ~25 °W, dominated by high values along the northern boundary. The mean contributions in the upper panel show three zonal peaks at 0 °E, 20 °W and 45 °W. These are related to the recirculation about the eastern sub-gyre, and the western sub-gyre respectively. The peak at 20 °W is particularly interesting as this is where the bottom bathymetry transitions from complex in the east to smooth in the west, which is known to impact diffusivity (Whalen et al., 2012) and overall circulation dynamics (Sonnewald et al., 2023).*

*The meridional mean in Fig. S8 also shows the closure of the mean advection component where the main heat source due to mean advection occurs in the southern limb (south of ~63°S) and the main heat sink due to mean advection occurs in the northern limb (north of ~63°S). Again, this agrees with all previous findings in the paper. Horizontal turbulent diffusion removes heat in the southern limb, and becomes a source of heat in the northern limb, with peaks occurring at ~61°S and ~59°S. These peaks might be related to the meridional change in the northern boundary across the area we are averaging over."*

[Figure]

Fig. 1. Map of total area for the figures below.

[Figure]

**Figure S7: upper panel: the zonal mean heat budget terms, in Wm⁻², for the whole Weddell Gyre west of 3 °E; lower panel: the corresponding cumulative heat budget terms in Terawatts (TW). The key for the legend is listed in Table 2. The dashed vertical line marks the approximate longitude of Maud Rise, at 3° E. The shaded errors provide the associated propagated error (detailed in section 3.2 and the supplement). The total region is marked by both blue and magenta stippling in Fig. 4.**

[Figure]

**Figure S8:** upper panel: the meridional mean heat budget terms, in Wm⁻², for the whole Weddell Gyre west of 3 °E; lower panel: the corresponding cumulative heat budget terms in Terawatts (TW). The key for the legend is listed in Table 2. The dashed vertical line marks the approximate longitude of Maud Rise, at 3º E. The shaded errors provide the associated propagated error (detailed in section 3.2 and the supplement). The total region is marked by both blue and magenta stippling in Fig. 4.

**Detailed comments:**

- Lines 25-: The plain language summary reads more like a second abstract. I think this needs to be more concise and less technical.

> We revised the summary as follows (Line 31):

> "***Plain Language Summary:*** *Ocean currents in the Weddell Sea are governed by a wind-driven clockwise circulating gyre, which is connected to the Antarctic Circumpolar Current to its north. Warm and salty deep water enters the Weddell Sea in its east, and is transported by the gyre circulation first southward, then westward, and back northwards again following the continental boundaries. During this circulation the water loses heat to the atmosphere and by contact with the ice shelves. Thereby the water becomes heavier, as also by salt released during the freezing of sea ice. The heaviest waters sink downwards along the Antarctic continental slope, eventually filling the deep abyssal ocean basins. The main source water mass for these processes and also the main source of heat to the Weddell Sea is called Warm Deep Water. Previous studies have shown the whole water column, especially in the deeper layers, is warming over recent decades in the Weddell Sea. The temperature of Warm Deep Water, however, fluctuates too strongly to tease out long-term trends from the "snapshot" data that is available to us. To better understand how heat is distributed in the Weddell Gyre within the Warm Deep Water, we combine temperature and velocity observations from a fleet of Argo floats freely drifting throughout the Weddell Gyre between 2002 and 2016. Using these observations, we estimate a heat budget in the layer that extends 1000 m deep from below the surface layer. This layer always includes the core of Warm Deep Water, regardless of its vertical position in the water column. Overall, large uncertainty prevents us from interpreting the results on a local scale, but interpretable features of heat flux divergence and convergence emerge when integrating the heat budget over large areas. The large-scale currents carry heat into the westward-flowing southern limb from the east, and upwelling brings heat upwards from below the layer throughout the whole gyre. Turbulent mixing, representing small scale processes, removes heat from the Warm Deep Water core through the top of the layer upwards into the ocean surface throughout. It also removes heat from the southern limb, northwards into the central gyre where Warm Deep Water recirculates and moves closer to the surface, as well as southwards towards the Antarctic coastline. Lastly, turbulent mixing also brings heat into the gyre across the northern boundary.*

- Line 34: "Warm Deep Water, however, varies in its properties too strongly to tease..." It is unclear what "varies too strongly" means.

> Replaced with (line 42):

> *"The temperature of Warm Deep Water, however, fluctuates too strongly to tease out long-term trends from the "snapshot" data that is available to us."*

- Line 39: "interesting features..." Please replace "interesting" with a more objective adjective.

> Replaced with (line 51): "*interpretable features of heat flux divergence and convergence*"

- Line 53: "The CDW that enters the Weddell Gyre is commonly referred to as Warm Deep Water (WDW)..." This is a nitpick, but I understand WDW to be a modified variant of CDW rather than simply CDW that exists in the Weddell Sea.

> Line 67: added the following *"becomes modified and is…"*

- Figure 1: Add contour labels for the streamlines, specifically the ones used to define IC SL subregions.

> We tried this, but the labels were not clear due to the figure already being quite busy. Instead, we include a new figure, figure 4, which clearly shows the IC and SL subregions, and the associated streamline labels (page 13).

- Line 152: Please briefly state how Sevellec et al. (2022) obtained their diffusivity estimates.

> Added the following (line 182):
>
> *"dataset provided by Sevellec et al. (2022), who derive horizontal diffusivities directly from Argo float trajectories by fitting a "pseudo-trajectory" to increase the spatial resolution required for the computation. Given this requires trajectory data without gaps in the record, estimates are missing for much of the Weddell Sea due to the presence of sea-ice."*

- Lines 135-138: A couple of things here:

- Figure S1a and a summary of the accompanying discussion regarding the definition of the vertical boundaries of WDW should be included in the main text.

> Done. Fig. 2 on page 6, along with the following text on line 152:
>
> *"This is to avoid incorporating highly seasonally variable surface waters from the analysis whilst also fixing the volume of water; (detailed explanation of the vertical boundaries is provided in the Supplements S1). Figure 2 shows selected vertical profiles with the upper and lower boundaries marked (the corresponding position of the profiles is found in Fig. S1, selected at random to provide a broad coverage of the Weddell Sea)."*

- For Figure S1a, it would be helpful to include additional profiles to illustrate the variability of temperature profiles and the location of the upper boundary.

> Done

- Regarding the previous point, are there regions where the lower boundary temperature is cooler than the upper boundary temperature?

> We checked all vertical profiles within the Weddell Sea, and found this not to be the case. The pale blue regions to the north of the Weddell Sea in the map of vertical advection (Fig. 3b), indicate where the deeper water is cooler than the upper boundary temperature. We added the following statement in Section 3.1:
>
> Line 156: *"Note the upper boundary temperature is always less than the lower boundary temperature within the Weddell Sea (there are regions where the opposite is true to the north of the gyre, within the ACC)."*

- Line 154: add "a" between acknowledging and lack.

> done

- Lines 203-205: Please provide a more physical motivation for defining the IC and SL regions. These seemingly arbitrary definitions undermine the robustness of these results.

> We have sought to clarify our motivations in the opening of Section 4.2 on Line 298, as detailed in response to your major comment referring to *"Over-reliance on arbitrary and ad hoc methods" on page 4 of this document.*

- Lines 254: I would rephrase "useful information" more objectively and state specifically why we should trust the spatially averaged values when the local details are not considered reliable.

> Changed the wording to the following on line 317:

> *"However, much of the local (grid-scale) imbalances (i.e., the random noise part) cancels out in the net (zonally integrated) heat budget terms, allowing regional patterns not affected by the differentiation at the grid scale to emerge"*

- Figure 3: Apologies if I missed this in the text, but what fraction of A_H goes north versus southward to the shelf?

> To compute this, we would need to directly compute advective heat fluxes along the streamline 26 and 16 Sv. We have not included this, since the results were rather noisy, and we assume the cross-stream flow is unlikely to be significant given the velocity estimates are derived from the stream function. This would be an important component to consider if we were able to look at deviations from the mean, which is unfortunately missing from the analysis, as you rightly pointed out. What we can say, from the lower panels in Figs 5-6 (before, Figs. 3-4), is where heat enters- and is removed from- a layer, and piece this information together by comparing the different sub-regions. We do, in Figs. 8-9, provide direct turbulent diffusive heat fluxes, since this is not dependent on the more spatially variable (and thus noisy) velocity variable, and thus provides less noisy results.

- Line 319: I would argue that the budget does not close anywhere in the domain.

> This sentence was deleted.

- Lines 323-324: It is odd to disregard the easternmost values in this section and not elsewhere. More consistency is needed. See my third major comment.

> The authors agree with you. We now provide the zonal analysis for regions west of 3°E, i.e., the longitude of Maud Rise, throughout. As a result the SL heat budget does very nearly come to a close (0.3 ± 3 TW, or 0.002 ± 0.02 °C/yr), although the IC still does not come to a close.

> We deleted this sentence and the subsequent 4 sentences as they are no longer relevant.

- Line 494: I am not sure what "ellipses" refer to.

> We have clarified this now (line 573):

> *"we hypothesised that a larger bias due to the horizontal gradient of the upper boundary depth (i.e., mid-thermocline, Fig. S1b) was occurring at the gyre periphery (i.e., where the slopes of the isopycnals are largest), which may be contributing to the large positive and negative values that extend diagonally outwards from the centre of the eastern sub-gyre (i.e. forming roughly shaped ellipses) in horizontal mean advection in Fig. 3a."*

- Lines 510-513: I suspect unresolved mesoscale eddies have a leading order impact on the mesoscale heat budget. In addition to the observational studies referenced in the following sentence, idealized modeling studies indicate that transient eddies (e.g., Wilson et al. 2022) are responsible for most of the southward heat transport in the eastern limb of the gyre.

> Thank you for bringing this up – the reference you cite is a really valuable addition. We have added the following sentence on line 601:

> "*Furthermore, Wilson et al. (2022) show, using idealised models, that transient eddies are responsible for most of the southward heat transport in the eastern limb of the Weddell Gyre.*"

> We altered the discussion from line 597, as detailed on page 2 of this document.

- Line 543: To be more precise, there is a Taylor Cap rather than a Taylor Column over Maud Rise. It is also inaccurate to say that the water column above the Rise is "stagnant" since it does exchange water mass properties with the ambient fluid.

> Thank you for pointing this out, we deleted the word "stagnant" and replace column with "cap".

- Figure 8: This is a lovely summary figure.

> Thank you! We have now added a second panel describing the features in the form of a legend, to shorten and simplify the caption.

**References:**

Ryan, S., Schröder, M., Huhn, O., and Timmermann, R.: On the warm inflow at the eastern boundary of the Weddell Gyre, Deep-Sea Res. I, 107, 70-81, https://doi.org/10.1016/j.dsr.2015.11.002, 2016.

Wilson, E. A., A. F. Thompson, A. Stewart, and S. Sun (2022), Bathymetric control of the subpolar gyres and overturning circulation in the Southern Ocean, Journal of Physical Oceanography, 52, 205–223, https://doi.org/10.1175/JPO-D-21-0136.1.

---

## Author Comment (AC3)

Modified figures for

**The Weddell Gyre heat budget associated with the Warm Deep Water circulation derived from Argo Floats**

Krissy Anne Reeve[1], Torsten Kanzow[1,2], Olaf Boebel[1], Myriel Vredenborg[1], Volker Strass[1], Rüdiger Gerdes[1,3]

[Figure]

**Figure 1: Sub-surface conservative temperature at the depth of temperature maximum ($\Theta_{max}$) with streamlines (grey contours) of the vertically integrated stream function for 50-2000 dbar with a spacing of 5 Sv, derived from in-situ observations from Argo floats (Reeve et al., 2019, 2016). Black dots show Argo float profile positions and red stars show mooring positions used in velocity field estimates. The thick black line shows the repeat ship-based transect from Kapp Norvegia to Joineville Island. The red circles labelled 1-3 show (1) Gunnerus Ridge, (2) Astrid Ridge and (3) Maud Rise. The black contours show the 1000, 2000 and 3000 m isobaths, from the general bathymetric chart of the oceans (GEBCO, IOC et al., 2003).**

[Figure]

**Figure 2: (a)** A random sample of vertical profiles of conservative temperature (the position of the profiles are marked by stars in Fig. S1b, the red star marking the position of the example profile in black), where Θmax is marked by a triangle, the upper boundary (i.e., mid-thermocline) and the lower boundary (mid-thermocline + 1000) are marked by squares, and the upper and lower WDW boundaries (defined by a neutral density range of 28 to 28.27 kg m-3 ) are marked by stars. This is to highlight that our method for the vertical boundary limits allows for the full inclusion of the core of WDW while also excluding Winter Water.

**Figure 3: The heat budget terms in W m⁻², from Eq. 1.2, for a layer of water 1000 m thick from the depth of the mid-thermocline: (a) mean horizontal geostrophic heat advection, (b) Total propagated error of mean horizontal advection, (c) mean vertical advection, (d) Total propagated error of mean vertical advection, (e) horizontal turbulent diffusion, (f) Total propagated error of horizontal turbulent diffusion (g) vertical turbulent diffusion, (h) Total propagated error of vertical turbulent diffusion, (i) the sum of the terms in a,c,e,g, and (j) the total propagated error of the sum of the terms. 
[revised manuscript text omitted]

[Figure]

**Figure S5: (a) Horizontal gradient in the upper boundary depth (i.e. mid-thermicline depth; m), and (b) horizontal gradient in the sub-surface temperature maximum (°C).**

[Figure]

**Figure S6: the cumulative sum of the diffusive heat flux in TW. (a) across the northern gyre boundary from west to east, where negative values indicate a southward flux of heat into the eastward-flowing northern limb of the Weddell Gyre from north of the northern Weddell Gyre boundary; (b) along the boundary between the southern inflow limb and IC-south, from east to west, where positive values indicate a removal of heat from the open southern limb of the gyre into the interior; in other words, a northward flux of heat from the southern limb into the interior circulation cell; (c) across the central gyre axis from IC-south to IC-north, where positive values indicate a diffusive heat flux northwards across the gyre axis. Errors are computed following the same method outlined in Section 3.2 and S7, with the cumulative net sum error summed in quadrature (i.e., the square root of the cumulative sum of the squared error).**

[Figure]

**Figure S7: upper panel: the zonal mean heat budget terms, in Wm⁻², for the whole Weddell Gyre west of 3 °E; lower panel: the corresponding cumulative heat budget terms in Terawatts (TW). The key for the legend is listed in Table 2. The dashed vertical line marks the approximate longitude of Maud Rise, at 3° E. The shaded errors provide the associated propagated error (detailed in section 3.2 and the supplement). The total region is marked by both blue and magenta stippling in Fig. 4.**

[Figure]

**Figure S8: upper panel: the meridional mean heat budget terms, in Wm⁻², for the whole Weddell Gyre west of 3 °E; lower panel: the corresponding cumulative heat budget terms in Terawatts (TW). The key for the legend is listed in Table 2. The dashed vertical line marks the approximate longitude of Maud Rise, at 3° E. The shaded errors provide the associated propagated error (detailed in section 3.2 and the supplement). The total region is marked by both blue and magenta stippling in Fig. 4.**

---

## Author Response (AR2)

**Response to reviewer 2 for EGU2023-21**

We thank the reviewer for their comments, and are happy to make the changes as suggested. The changes are provided in response to each comment (in blue) listed below, with line numbers referring to the manuscript with tracked changes:

- Line 34 (plain language summary): This explanation of CDW transformation is incomplete. Some of this water mass becomes lighter by mixing with surface waters.

Line 34: "Warm and salty deep water enters the Weddell Sea in its east, and is transported by the gyre circulation first southward, then westward, and back northwards again following the continental boundaries. During this circulation, some of this water becomes lighter by mixing with the overlying surface waters, thus shoaling as it circulates. It also loses heat to the atmosphere and by contact with the ice shelves. When this occurs, the water becomes heavier, as also by salt released during the freezing of sea ice."

- Line 171: The unresolved time variability term is also unknown.

Line 177: "Thus, in addition to the unknown time variability term, $d\Theta/dt$, we also have an unknown 5$^{th}$ term in the heat budget in Eq. 1.1 and 1.2, R"

- Line 218: Sorry for not mentioning this earlier, but is this double-gyre a robust circulation feature? This is the first time I have encountered a description of a sub-gyre in the eastern Weddell.

Reeve et al. (2019) is the first comprehensive study showing the synoptic scale horizontal circulation from in situ measurements, which shows the double-cell structure of the Weddell Gyre. It has also been shown in numerical model simulations previous to this work (Beckmann et al., 1999; Timmermann et al., 2002), and hypothesized in early observation-based studies (with the spatial limitations of observations preventing more than speculation): Matano et al., (2002) suggested the cause is topographic steering where the SW Indian Ridge constrains the flow in the northern limb east of the Prime Meridian, and Orsi et al., (1993) observed that the double-cell structure is more prominent in deeper layers, which may be why it has been challenging to observe. More recent studies providing estimates of volume transports across various ship-based transects in the Weddell Sea also support the concept of a double-gyre structure, in that the eastern sub-gyre exhibits stronger circulation (and thus transport estimates) than in the western Weddell Sea (see figure 7 in Reeve et al. 2019). The lead author has preference for the terminology of "sub-gyres", because it is not clear how significant, or how synced, these sub-gyres are, and there also appears to be a network of sub-gyres in play across the entire Southern Ocean (Sonnewald et al., 2023).

We added the following sentence in the opening to section 4, line 239:

"While large uncertainties are associated with the eastern part of the eastern sub-gyre, numerical model simulations (e.g., Timmermann et al., 2002), historical observations (e.g., Orsi et al. 1993), and direct volume transport estimates support the concept of a double-gyre structure in the Weddell Sea (e.g., Fig. 7 in Reeve et al., 2019)."

- Line 270: Sorry if I missed this, but it would be helpful to state why the analysis focuses on the region west of Maud Rise (i.e. because the uncertainty is too great in the eastern Weddell).

We explain this in the results section part 2, when integrating zonally. Line 312:

"For both regions, the area east of Maud Rise (3° E) is omitted, due to large uncertainties east of Maud Rise (discussed in Section 5)."

- Line 475: According to Table 3, the heat budget for the IC region does not close.

We updated the sentence accordingly (now line 539), and checked the rest of the document for any related sentences in need of updating. Thanks for pointing this out (it was remnant of the results before we updated the analysis).

"While the heat budget does not close on regional scales, it does approximately close when integrating over the open southern limb west of Maud Rise (SL), but not over the interior west of Maud Rise (IC). Nonetheless, important and useful information can be provided from comparing the four resulting heat budget terms."

**References**

Reeve, K. A., Boebel, O., Strass, V., Kanzow, T., Gerdes, R.: Horizontal Circulation and volume transports in the Weddell Gyre derived from Argo float data, Progress in Oceanography, 175, pp. 263-283, https://doi.org/10.1016/j.pocean.2019.04.006, 2019.

Beckmann, A., Hellmer, H.H. and Timmermann, R.: A numerical model of the Weddell Sea: Large-scale circulation and water mass distribution. Journal of Geophysical Research: Oceans, 104(C10), pp.23375-23391, 1999.

Timmermann, R., Beckmann, A., and Hellmer, H. H.: Simulations of ice-ocean dynamics in the Weddell Sea 1. Model configuration and validation, J. Geophys. Res., 107( C3), doi:10.1029/2000JC000741, 2002.

Matano, R.P., Gordon, A.L., Muench, R.D. and Palma, E.D.: A numerical study of the circulation in the northwestern Weddell Sea. Deep Sea Research Part II: Topical Studies in Oceanography, 49(21), pp.4827-4841, 2002.

Orsi, A.H., Nowlin, W.D., Whitworth, T.: On the circulation and stratification of the Weddell Gyre, Deep Sea Research Part I: Oceanographic Research Papers, Volume 40, Issue 1, Pages 169-203, ISSN 0967-0637, https://doi.org/10.1016/0967-0637(93)90060-G, 1993.

Sonnewald, M., Reeve, K.A. & Lguensat, R.: A Southern Ocean supergyre as a unifying dynamical framework identified by physics-informed machine learning. Commun Earth Environ 4, 153, https://doi.org/10.1038/s43247-023-00793-7, 2023.